# Dietary nutrients of relative importance associated with coronary artery disease: Public health implication from random forest analysis

**Til Bahadur Basnet**[1]\*, **Srijana G. C.**[2], **Rajesh Basnet**[3], **Bidusha Neupane**[3]

**1** Little Buddha College of Health Sciences, Prubanchal University, Kathmandu, Nepal, **2** Maharajgunj Nursing Campus, Tribhuvan University, Kathmandu, Nepal, **3** Institute of Medicine, Tribhuvan University, Kathmandu, Nepal

\* ddst19basnet@hotmail.com

**Data Availability Statement:** All relevant data are within the manuscript and its Supporting Information files.

## Abstract

Dietary nutrients have significant effects on the risk of cardiovascular diseases. However, the results were not uniform across different countries. The study aims to determine the relative importance of dietary nutrients associated with coronary artery disease (CAD) among the Nepalese population. A hospital-based matched case-control study was carried out at Shahid Gangalal National Heart Center in Nepal. In the present study, patients with more than seventy percent stenosis in any main coronary artery branch in angiography were defined as cases, while those presenting normal coronary angiography or negative for stressed exercise test were considered controls. Dietary intakes of 612 respondents over the past 12 months were evaluated using a semi-quantitative customized food frequency questionnaire. In conditional regression model, the daily average dietary intake of β-carotene (OR: 0.54; 95%CI: 0.34, 0.87), and vitamin C (OR: 0.96; 95%CI: 0.93, 0.99) were inversely, whereas dietary carbohydrate (OR: 1.16; 95%CI: 1.1, 1.24), total fat/oil (OR: 1.47; 95%CI: 1.27, 1.69), saturated fatty acid (SFA) (OR: 1.2; 95%CI: 1.11, 1.3), cholesterol (OR: 1.01; 95%CI: 1.001, 1.014), and iron intakes (OR: 1.11; 95%CI: 1.03, 1.19) were positively linked with CAD. Moreover, in random forest analysis, the daily average dietary intakes of SFA, vitamin A, total fat/oil, β-carotene, and cholesterol were among the top five nutrients (out of 12 nutrients variables) of relative importance associated with CAD. The nutrients of relative importance imply a reasonable preventive measure in public health nutrients specific intervention to prevent CAD in a resource-poor country like Nepal. The findings are at best suggestive of a possible relationship between these nutrients and the development of CAD, but prospective cohort studies and randomized control trials will need to be performed in the Nepalese population.

**Funding:** The author(s) received no specific funding for this work.

**Competing interests:** The authors have declared that no competing interests exist.

## Introduction

Coronary artery disease (CAD) is a significant cause of disability and premature death throughout the world. The underlying pathology is atherosclerosis, which develops over many years and is usually advanced by the time symptoms occur, generally in middle age [1]. An estimated 7.4 million people died from CAD in 2015, representing 13% of all global deaths [2]. Primary risk factors are tobacco use, unhealthy diet, physical inactivity, harmful alcohol consumption. These, in turn, show up in people as raised blood pressure, elevated blood glucose, and overweight and obesity risks detrimental to good heart health [3].

A global strategy based on knowledge of the importance of risk factors for cardiovascular disease (CVDs) in different geographic regions and various ethnic groups is needed to prevent diseases effectively [4]. Western people mostly rely on high energy-dense food, substantially not reducing the incidence of obesity and CVDs [5] despite improved medical care and an increase in the cessation of smoking [6]. In low and middle-income countries, fast food, energy-dense food, and diet in high fat are related to an abrupt rise in CAD even in a younger population of high socioeconomic status of an urban area [7].

Dietary habit is considered as one of the potentially modifiable risk factors for CVDs. The quality of dietary carbohydrates plays a significant impact on the development of metabolic diseases and CAD. For instance, refined sugar increases CAD's risk, while complex carbohydrates lower CAD incidence [8, 9]. Total dietary fat was associated with an increased risk of CVD and all-cause death [10]. After adjustment of some coronary heart disease (CHD) risk factors, higher intakes of polyunsaturated fatty acids (PUFA) and monounsaturated fatty acids (MUFA) were associated with a reduced risk of CHD [11]. Unlike observational studies, randomized control trials suggest that SFAs either do not or only modestly increase the risk for CAD [12, 13]. Evidence supports that industrially-produced trans-fats are linked to increased risk for CVDs [12, 14]. Thus, dietary fat quality contributes to the risk of the leading chronic diseases and is more critical than the quantity of fat/oil in reducing the risk of chronic disease mortality, especially from CVDs.

Dietary fat is rice in energy sources; it also carries fat-soluble vitamins and other nutritious substances, provides essential fatty acids, and aids physiological functions in the body [15]. PUFAs have been of great interest for human health due to their potential anti-inflammatory action that may protect from several chronic-degenerative diseases with inflammatory pathogenesis [16]. In many countries, daily intake of saturated fats exceeds the recommended limit of 10% energy (%E), while intakes of polyunsaturated fats (PUFAs) are often below the recommended range of 6–11%E, and consumption of long-chain ω–3 PUFAs is exceptionally low [17]. The average intake level of fat and carbohydrate varies in different countries, regions, and groups of people across the world. Thus, dietary fat recommendations must consider each country's dietary fat/oil patterns [17]. However, there are discrepancies in the research findings of the role of vitamin B and antioxidant vitamins for CAD development [18–21].

Like in high-income countries, major traditional risk factors: tobacco use, alcohol consumption, unhealthy diet, and physical inactivity were reported in prevalence studies in the Nepalese population [22]. Although people's food habits vary with different ethnicities and geography, Nepalese commonly consume rice or bread, pulses, and vegetables, with potatoes prepared mostly in vegetable oil as the main meal. Nonetheless, the daily intakes of animal products, fruits, and vegetables are low in Nepalese [23]. However, analytical studies to determine the strength of association between dietary nutrients and CVDs were scarce in the Nepalese population. Therefore, the present study was carried out to examine the association of energy intake, dietary macronutrients, and micronutrients with CAD.

## Materials and methods

A matched case-control analytical study was designed to describe the relation of nutrients to CAD. The data of 612 participants were collected from June 4 to September 4, 2018. The face to face interview was accomplished with patients who visited at highly specialized central-level cardiac treatment center, Shahid Gangalal National Heart Center in Nepal, which provides curative services to patients with CVDs. An ethical review board of the Nepal Health Research Council (308/2017-18) approved the present study. Before conducting a survey, permission was taken for collecting clinical data from the respective hospital. Written consent was taken from the respondent before attending the interview.

### Study participants

Study cases were selected from admitted patients after suspected myocardial infarction or exercise-induced stress test positive or from those who would electively undergo angiography in the hospital. After angiography, patients with stenosis higher than 70 percent (%) in any main coronary artery branch were defined as study cases. The controls were patients who were either presenting normal coronary angiography or those who were negative for stressed exercise test, also called a treadmill exercise tests (TMT). TMT was carried out for patients who were essentially referred by a physician for the test based on a chief complaint of chest pain and or at least one cardiometabolic risk factor (either hypertension or dyslipidemia or diabetes). Those patients who visited the hospital for an elective check-up or whole body check-up, including TMT test, were also included in the TMT test. A case to control was 1:1; altogether, 306 cases and 306 controls were included in the study. "Sex" and "Age" were matched at the individual level and an interval of five years, respectively. Patients having a report with aortic valve sclerosis on echocardiogram and any abnormality on electrocardiogram were excluded. Severely ill patients like kidney failure, cancer, and heart failure were not included. The participant selection flowchart was presented in Fig 1.

### Data collection technique and tools

Data was collected through face to face interviews and observation, using a semi-structured food frequency questionnaire (FFQ) tool and observation checklist, respectively. The questionnaire set comprised of three-part, namely i. General socio-demographic characteristic ii. Cardio-metabolic and behavioral risk factors and iii. Food frequency questionnaire. Blood pressure, weight and height, waist and hip circumference of cases and controls were measured and recorded using the observation checklist. Data for cases were collected on the second day of angiography after patients became stable. The average time for an interview in FFQ for cases and control was 42±7 minutes and 39±6 minutes, respectively.

### Assessment of covariates

A standardized protocol was used to measure the height, weight, and waist and hip circumferences. A wall-mounted stadiometer measured height to the nearest centimeter. We asked respondents to stand upright without shoes, with their back against the wall, heels together, and eyes directed forward. Their weight was measured with a portable electronic weighing scale placed on a firm horizontal surface. Waist and hip circumferences were measured with a non-stretchable standard measuring tape. Waist measurements were obtained over a lightly clothed abdomen at the narrowest point between the costal margin and iliac crest, and hip circumference was measured over light clothing at the level of the widest diameter around the buttocks. Body mass index (BMI) was categorized as normal ($<23.0$ kg/m$^2$), overweight (23.0

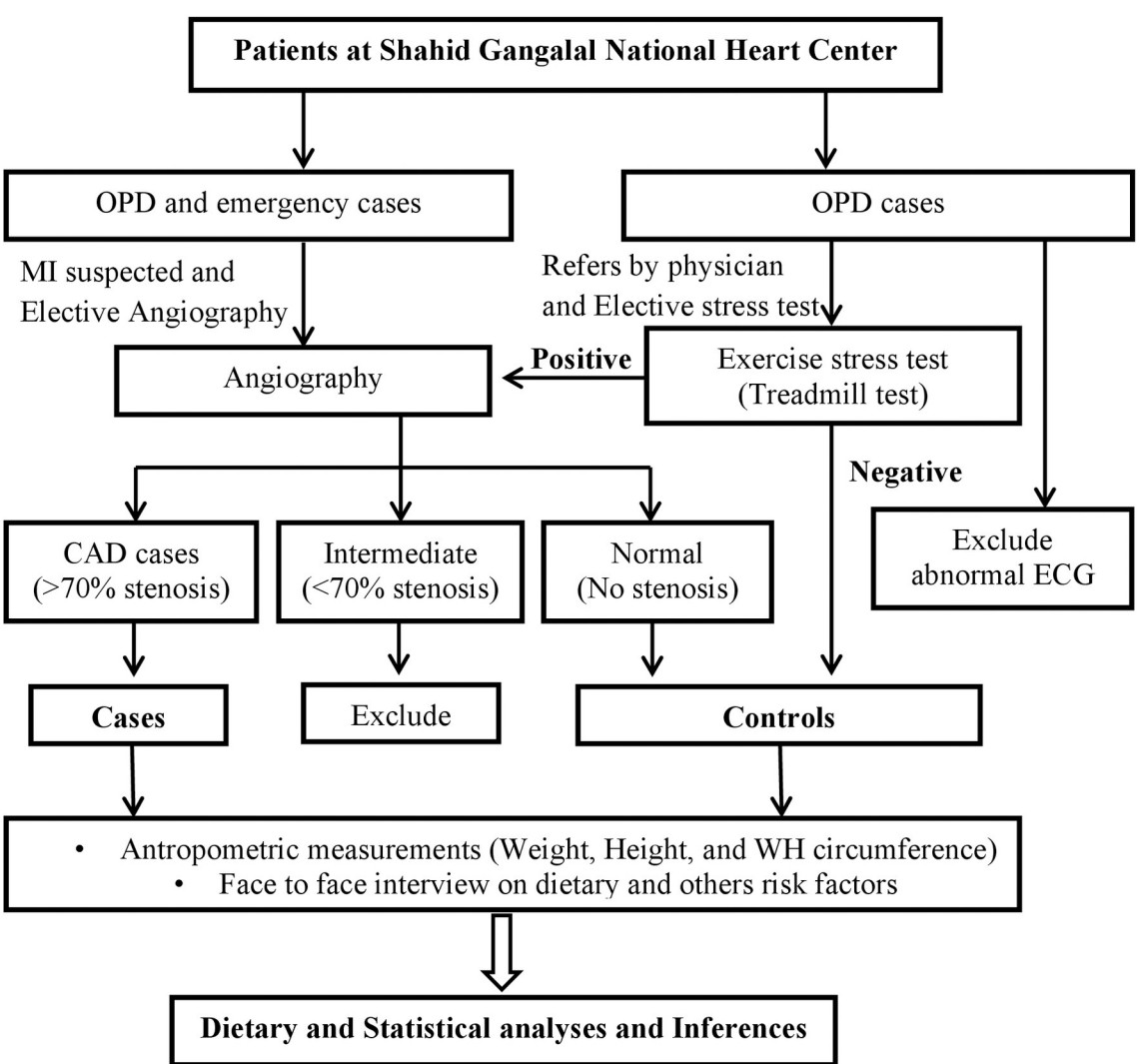

**Fig 1. Flowchart diagram for selection of study participants for cases and controls.** OPD: Outpatient department; CAD: Coronary artery disease; ECG: Electrocardiogram, WH: Waist hip.

to $< 27.5.0$ kg/m$^2$), and obese ($\geq 27.5.0$ kg/m$^2$) [24]. Abdominal obesity was defined as waist circumference $\geq 90$ centimeters in males and $\geq 80$ centimeters in females [25]. Dyslipidaemia was defined as hypercholesterolemia or hypertriglyceridemia; if the high-density lipoprotein (HDL) level was below 30 mg/dl, dyslipidemia was considered. Hypertriglyceridemia and hypercholesterolemia were defined as triglyceride (TG) serum and total cholesterol (TC) levels greater than 150 and 200 mg/dl, respectively, or if hypolipidemic treatment was administered [26]. Diabetic individuals were those with fasting blood glucose equal to or greater than 126 mg/dl throughout two tests or those taking diabetes medications [26]. Patients whose blood pressure was greater or equal to 140/90 mmHg or those taking antihypertensive medication according to their medical records were classified as hypertensive [27].

Persons who smoked until within one year of the interview were considered current daily smokers. Never smokers were those who responded with "occasionally" or "not at all" on the questionnaire, and ex-smokers were those who smoked daily before one year of interview. Current alcohol drinkers were categorized as those who engaged in alcohol drinking within

the last year. A total of 13.6 g of pure alcohol was considered one standard drink, equivalent to consuming 43 ml of local alcohol (Raksi) and 341 ml of beer, Zaand, or Tongba [28].

Physical activity levels related to work were categorized as vigorous, moderate, or low. Vigorous physical activity was considered any activity that caused a substantial rise in heart and breathing rates, such as digging or plowing fields, lifting heavy weights, etc. Continuous engagement in such activity for at least 10 minutes was considered involvement in vigorous activity. Similarly, moderate physical activity was defined as any activity that caused a moderate increase in heart and breathing rates (examples include domestic chores, gardening, lifting light weights, etc.). Continuously engaging in such activity for at least 30 minutes was considered involvement in moderate activity. Physical activity related to transport was not considered in this study. The recreational activity was also called physical exercise which included two types of activities, vigorous and moderate, based on exertion. The vigorous recreational activity was defined as any recreational activity that significantly increased heart and breathing rates, such as football, fast swimming, and rapid cycling. Ten minutes of such activity was considered involvement in vigorous recreational activity. The moderate recreational activity was defined as any recreational activity that causes a moderate increase in heart and breathing rates; examples include yoga, playing basketball, brisk walking, and regular cycling [29]. During analysis, total physical activity (related to work and recreational) were categorized as "Yes" (includes moderate and vigorous) and "No."

## Dietary assessment

The FFQ was semi-structured, modified, and validated European Prospective Investigation of Cancer FFQ, customized to Nepalese day to day food items for obtaining detailed information regarding dietary nutrients and edible fat and oil intake from study participants. The list of fifty-nine food items (S1 Table), which were frequently consumed by the Nepalese population, was included. For testing internal validity, and intra-class correlation analyses were performed between FFQ1 and FFQ2, criterion validity was measured by Pearson's correlation of FFQ2 with a "24-h recall diet survey" as a gold standard (S2 Table). An average of at least three days (two weekdays and one weekend) 24-h dietary recall can estimate about mean dietary intake for a day on the population or large-group level [30]. It is an intensive method for assessing dietary intake and is commonly used as a comparison method for validation/calibration studies of structured assessments such as FFQ [31]. However, it does not accurately determine the usual intake over time due to the large intra-individual variation in dietary intakes [30]. As FFQ can provide important information about dietary patterns, it is the most commonly used instrument to assess past dietary intake in epidemiological studies on the relationship between dietary factors and diseases [31].

Intake frequencies for the food items consisted of nine categories ranging from never to more than six times per day. Participants were asked how often they had consumed each food item listed on the FFQ during the past year. After being diagnosed as having any one of CAD's risk factors (obesity/hypertension/diabetes/dyslipidemia), patients who modified their dietary habits and those taking dietary supplements and vitamins were excluded from the study. The quantity of each food consumed by a subject was calculated by multiplying its consumption frequency by the usual amount consumed. Dietary intakes were then calculated using a programmed nutrition calculator based on the value of nutrients per 100 g food consumed per day for each food item with the use of the Nepal food composition table 2017.

A food atlas with color photographs of three portions sized- small, medium, and large- for various food items was developed and displayed to respondents to minimize the recall bias. The food items that did not have natural units or applicable household measures were

photographed. Different sizes of glasses or bowls were displayed to estimate the number of liquids. Other items were asked as several specified units (slice, number, spoon, etc.). As people usually consumed seasonal fruits and vegetables, we grouped them into "leafy vegetables" and "other vegetables." Several vegetables are cooked in combination, including potato, onion, and tomato. Some types of vegetables are available in only one season, while others are consumed more than one season. To address the overestimation of these above problems, we calculated average nutrients per 100 g that were allotting more weight to those vegetable items (cauliflower, mustard leaves, tomatoes), which are consumed more extended periods. Liquid oils are used during food preparation, so it is not easy to estimate the actual amount of oil consumed for an individual. Therefore, we asked separately the average number of days that would be sufficient to cook the foods from one liter of oil or ghee. Then, we calculated the total average amount of oil intake per person per day. Finally, fifty-nine food items were grouped into twenty-five food items with specific food ratios from Nepal's food composition table and translated into particular dietary nutrients value.

## Statistical analyses

Firstly, the Mcnemar test for categorical variables and Wilcoxon sign ranked test for continuous variables were executed to find the association between variables and CAD. After that, a conditional logistic regression model was constructed to test the strength of the association between dietary nutrients and CAD. By stepwise backward deletion process, we developed an energy-adjusted parsimonious model. To adjust the collinearity problem among our data variables, we performed random forest (RF) analysis, explaining the nutrients variability associated with CAD. RF consists of many individual de-correlated decision trees by sampling the random set of original data operating as an ensemble. Each tree gives a classification, and the forest selects the classification having the most votes across all the trees in the forest. RF is also common to perform the prediction task in the medical domain [32, 33]. The receiver operating characteristic (ROC) curve was constructed to assess the logistic regression and RF performance. Data analysis was performed with R packages in version 3.6.2, and two-tailed tests with $p$-value $<0.05$ were considered significant.

## Results

Proportions of socio-demographic, cardio-metabolic, and behavioral characteristics of respondents between cases and controls in the study are presented in Table 1. The median age was 58 years, which was the same in both case and control groups. Because "Age" was matched at an interval of five years in the study, it was significantly associated with CAD ($p$-value = 0.001) in bivariate analysis. As CAD incidence is lower in the female population, fewer female CAD cases were admitted to the hospital, resulting in the ratio of male-to-female cases of 3:1. Similarly, diabetes mellitus, dyslipidemia, and general and abdominal obesity were significantly related to CAD ($p$-value $<0.001$). Regarding behavioral factors, smoking ($p$-value $< 0.001$), alcohol use ($p$-value = 0.019), and physical activity ($p$-value $< 0.001$) were found to be statistically significant linked with the disease.

As the data of most of the variables were not normally distributed, median and interquartile values of nutrients intake are presented (Table 2). The median energy intake per day was 2674 (2445, 2909) and 2622 (2373, 2963) in control and case groups showing no statistically significant difference ($p$-value = 0.679). Total fat/oil, fiber, vitamin C, beta (β)-carotene, vitamin A, MUFA, SFA, and cholesterol intake per day were showing significant association with CAD ($p$-value $< 0.001$). In contrast, carbohydrate and PUFA intakes were significantly linked with CAD ($p$-value = 0.002). Also, a significant association of dietary zinc ($p$-value = 0.023), iron ($p$-

**Table 1. Socio-demographic, cardio-metabolic, and behavioral characteristics of respondents between cases and controls in the study.**

| Variables | | Control (%) n = 306 | Case (%) n = 306 | p-value[a] |
|---|---|---|---|---|
| **A. Demographic risk factors** | | | | |
| **Age** | | 58 (50, 65)[b] | 58 (50, 65) | <0.001*** |
| **Sex** | Female | 73 (23.9) | 73 (23.9) | 1.000 |
| **B. Cardiovascular Risk factors** | | | | |
| **Diabetes** | Yes | 40 (13.1) | 86 (28.1) | <0.001*** |
| **Hypertension** | Yes | 143 (46.7) | 142 (46.4) | 0.938 |
| **Dyslipidemia** | Yes | 32 (10.5) | 90 (29.4) | <0.001*** |
| **General obesity** | Yes[c] | 63 (20.6) | 134 (43.8) | 0.001*** |
| **Abdominal obesity** | Yes[d] | 160 (52.3) | 240 (78.4) | <0.001*** |
| **C. Behavioral Risk Factors** | | | | |
| **Alcohol use** | Yes | 66 (21.6) | 91 (29.7) | 0.019* |
| **Smoking** | Never | 196 (64.1) | 113 (36.9) | <0.001*** |
| | Ex-smoker | 70 (22.9) | 73 (23.9) | |
| | Current smoker | 40 (13.1) | 120 (39.2) | |
| **Physical activity** | Moderate/ vigorous | 241 (78.8) | 202 (66) | <0.001*** |

[a]Mcnemar test (categorical variables) and Wilcoxon sign ranked test (continuous variables).

[b]Median and interquartile value.

[c]General obesity: Body mass index $\geq$ 27.5 kg/m$^2$ (for Asian).

[d]Abdominal obesity: Waist hip ratio $\geq$ 0.8 for females and $\geq$ 0.95 for males (for Asian).

*$p \leq .05$

**$p \leq .01$

***$p \leq .001$.

value = 0.003), thiamine (p-value = 0.013), and riboflavin (p-value = 0.016) with CAD was reported. However, dietary intakes of protein, calcium, phosphorous, and niacin were not significantly associated with the disease.

Those micronutrients that showed potential associations with CAD (p-value < 0.05) (Table 2) were then tested in the conditional logistic regression analysis (Table 3). In this analysis, we developed a parsimonious energy-adjusted model by stepwise backward deletion. The final model was adjusted with dyslipidemia, diabetes mellitus, smoking, and BMI. In this model, those nutrients inversely related to CAD were β-carotene (OR: 0.54; 95%CI: 0.34, 0.87) and vitamin C (OR: 0.96; 95%CI: 0.93, 0.99) indicating possible protective factors. However, dietary carbohydrate (OR: 1.16; 95%CI: 1.1, 1.24), total fat/oil intake (OR: 1.47; 95%CI: 1.27, 1.69), SFA (OR: 1.2; 95%CI: 1.11, 1.31), cholesterol (OR: 1.01; 95%CI: 1.001, 1.014), and iron intakes (OR: 1.11; 95%CI: 1.03, 1.19) were shown proportionately linked with CAD indicating probable risk factors.

We performed random forest analysis to evaluate the important dietary nutrients linked with CAD. In this RF model, we incorporated all the variables in the first step of conditional logistic regression analysis. The top twelve variables were presented in Fig 2; the five topmost important dietary nutrients linked with CAD were SFA, vitamin A RE, total fat/oil, β-carotene, and cholesterol. In this RF analysis, 250 trees and four variables were tried in each split, where the out-of-bag (OOB) estimate of error rate was 16%.

Fig 3 reveals the area under curve (AUC) that evaluated the logistic regression model and RF model. Although the RF model had a lower AUC (90%) and other performance parameters (Table 4) than the logistic model (AUC—96%), RF adjusted the effect of multi-collinearity so that the interaction between correlates was independent.

**Table 2. Distribution of nutritional factors associated with coronary artery disease between case and control groups in the study.**

| Nutrients intake/day | Control (n = 306) | Case (n = 306) | p-value[a] |
|---|---|---|---|
| Food energy Kcal | 2674 (2445, 2909)[b] | 2622 (2373, 2963) | 0.679 |
| Protein g | 70 (61, 78) | 67 (59, 77) | 0.169 |
| Total fat/oil g | 56 (47, 64) | 61 (52, 72) | <0.001*** |
| Carbohydrate g | 433 (388, 481) | 409 (368, 456) | 0.002** |
| Fiber g | 11.2 (9.8, 12.9) | 10.1 (8.7, 11.6) | <0.001*** |
| Calcium mg | 683 (434, 929) | 648 (363, 930) | 0.244 |
| Phosphorus mg | 1431 (1287, 1639) | 1407 (1210, 1615) | 0.146 |
| Iron mg | 21.3 (18.6, 24) | 20 (17.4, 23.5) | 0.003** |
| Zinc mg | 14.1 (12.2, 16.2) | 13.8 (11.7, 15.5) | 0.023* |
| Thiamine mg | 1.2 (1, 1.4) | 1.2 (0.97, 1.3) | 0.013* |
| Riboflavin mg | 1.1 (0.89, 1.3) | 1.04 (0.79, 1.3) | 0.016* |
| Niacin mg | 13.6 (11.9, 16.6) | 13.8 (11.4, 16) | 0.671 |
| Vitamin C mg | 45.3 (38, 52.9) | 39.5 (33, 48.7) | <0.001*** |
| β-carotene mcg | 2579 (2162, 3352) | 2227 (1863, 2698) | <0.001*** |
| Vitamin A R.E. | 698 (546, 836) | 622 (506, 728) | <0.001*** |
| PUFA g | 18.7 (12.5, 23.5) | 19.6 (12.4, 25.7) | 0.002** |
| MUFA g | 17.2 (12.4, 23) | 19.2 (14, 25.6) | 0.001*** |
| SFA g | 15.5 (10.6, 19.2) | 19 (13.9, 23.6) | <0.001*** |
| Cholesterol mg | 108 (71,163) | 129 (94,181) | <0.001*** |

Kcal: kilocalorie; g: gram; mg: milligram; mcg: microgram; R.E.: retinol equivalent; PUFA: polyunsaturated fatty acid; MUFA: monounsaturated fatty acid; SFA: saturated fatty acid.

[a]Wilcoxon sign ranked test.

[b]Median and interquartile value.

*p ≤ .05

**p ≤ .01

***p ≤ .001.

## Discussion

The present hospital-based matched case-control study was designed to determine the association of dietary nutrients with CAD in the Nepalese population. Dietary intakes of

**Table 3. A model based on conditional logistic regression analysis showing the effect of nutrients intake on coronary artery disease.**

| Nutrients intake/day | OR (95%CI) | p-value |
|---|---|---|
| Carbohydrate g | 1.16 (1.1, 1.24) | <0.001*** |
| Total fat/oil g | 1.47 (1.27, 1.69) | <0.001*** |
| SFA g | 1.2 (1.11, 1.31) | <0.001*** |
| Cholesterol g | 1.01 (1.001, 1.014) | 0.016* |
| β-carotene mg | 0.54 (0.34, 0.87) | 0.011* |
| Vitamin C mg | 0.96 (0.93, 99) | 0.018* |
| Iron mg | 1.11 (1.03, 1.19) | 0.005** |

g: gram; mg: milligram; SFA: saturated fatty acid; OR: odds ratio; CI: confidence interval.

*p ≤ .05

**p ≤ .01

***p ≤ .001.

The model was adjusted daily energy intake and with cardio-metabolic risk factors (Diabetes mellitus (type II), dyslipidemia and body mass index), and smoking.

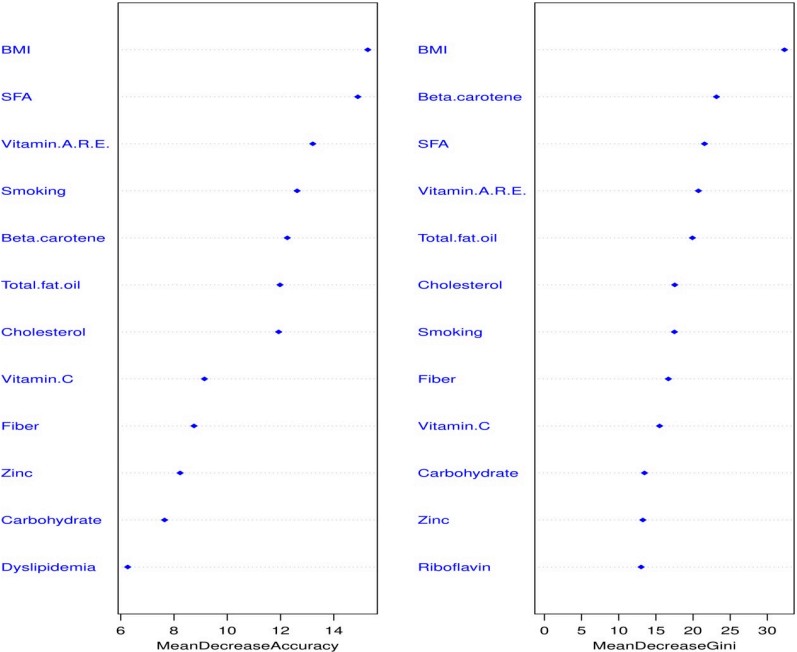

**Fig 2. The essential nutritional and traditional variables in random forest regression analysis.** WH ratio: Waist-Hip ratio; g: gram; T2DM: Type II Diabetes mellitus; mg: milligram; mcg: microgram; RE: Retinol equivalent; PUFA: polyunsaturated fatty acid; MUFA: monounsaturated fatty acid; SFA: saturated fatty acid; kcal: kilocalorie.

carbohydrate, total fat, fiber, riboflavin, thiamine, vitamin C, β-carotene, vitamin A, vitamin C, PUFA, MUFA, SFA, and cholesterol were significantly associated with CAD in bivariate analysis. Furthermore, in multivariable conditional logistic regression analyses, we developed an energy-adjusted parsimonious model by stepwise backward deletion process where we adjusted three cardio-metabolic factors (diabetes, dyslipidemia, and BMI) and smoking.

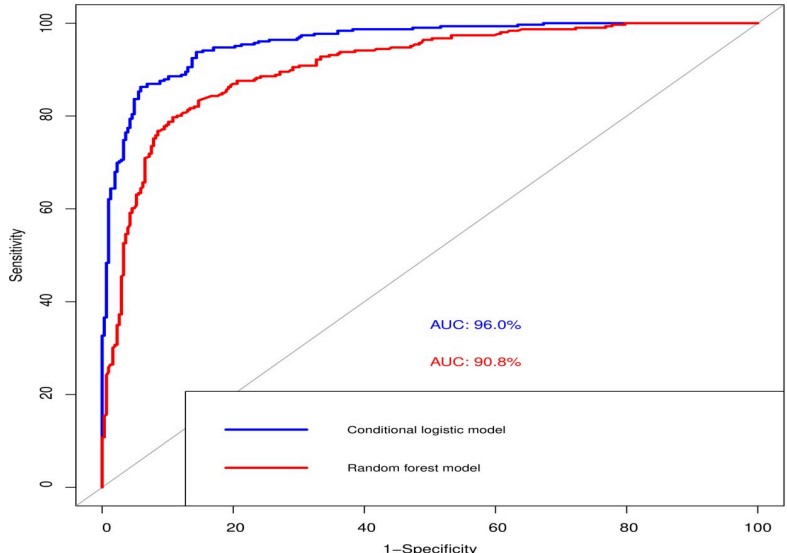

**Fig 3. Receiver operating curve (ROC) to compare the logistic regression model and in the random forest model.** AUC: area under curve.

**Table 4. Comparison of performance for logistic regression and random forest.**

| Characteristics | Logistic regression | Random forest |
|---|---|---|
| Accuracy | 0.902 | 0.843 |
| Positive predictive rate | 0.936 | 0.84 |
| Negative predictive rate | 0.873 | 0.846 |
| No. of true positive/false negative | 264/18 | 257/49 |
| No. of true negative/false positive | 288/42 | 259/47 |
| Sensitivity | 0.863 | 0.845 |
| Specificity | 0.941 | 0.841 |
| F1 score | 0.898 | 0.843 |
| Out of bag error estimate (%) | - | 15.69 |

However, even though hypertension is a well-established risk factor, we did not observe a significant association; therefore, we did not include it in the model construction. The reason for the insignificant association of hypertension might be selecting the control groups from the same hospital's outdoor patients, where more hypertensive patients come for their heart check-up. In the final model, we observed that dietary β-carotene and vitamin C were inversely associated with CAD, whereas higher dietary carbohydrate, total fat and oil, SFA, cholesterol, and iron intake were directly associated with CAD. We also performed random forest analysis to adjust the collinearity problem and then identify the topmost nutritional variables. Random forest analysis revealed that dietary intake of SFA, vitamin A, total fat and oil, β-carotene, and cholesterol were the topmost important nutrients associated with CAD. Even though dietary vitamin A was not significantly associated with CAD in conditional regression analysis, it remained important nutrients related to CAD in the random forest analysis.

The quantity and type of dietary fat/oil and carbohydrate consumption, fiber, protein, and alcohol intakes strongly impact blood lipids and lipoprotein metabolism, thereby developing CVDs [34]. Energy from complex carbohydrates has many benefits compared to refined sugars [34]. For example, a low glycemic index diet could improve blood lipids and blood pressure [35], thus reducing CAD [36]. Short term carbohydrate diet reduces the weight and atherosclerotic plaque in CAD, but the long term effect is still controversial [37]. Besides, a low carbohydrate diet is not associated with coronary artery incidence and progression [38]. The present study showed that an increased intake of dietary carbohydrates had a risk of CAD. However, we were unable to differentiate the dietary carbohydrate into refined and complex carbohydrates. A higher intake of fat without replacing protein and carbohydrates causes metabolic disorders related to CAD [39]. We found that total fat intake was proportionately associated with increased risk of CAD. However, dietary fat types, but not total fat intake, are an important determinant of CVDs [40]. Specifically, intakes of PUFAs and MUFAs are associated with a lower risk of CVDs and death, whereas SFA and trans-fat intakes are linked with a higher risk of CVDs [39]. Consistently, we reported significantly higher odds of CAD with increased SFA intake; PUFA and MUFA intakes were inversely but not significantly associated with CAD. The higher intake of carbohydrates, total fat, and SFA indicated an unhealthy eating pattern among the Nepalese population that might increase CAD prevalence.

Nevertheless, previous literature demonstrated a discrepancy in results among these specific fat intake groups. For example, SFA is proportionately associated with CAD [41], but not in the Kuopio Ischemic Heart Disease Risk Factor Study [42] and meta-analysis [43]. Likewise, in a recent meta-analysis, dietary intake of PUFA was not found significantly associated with CAD [44]. Besides, cholesterol is also independent risk factors of CAD according to lipid theory [45]. Incongruous with this study, we also reported a significant relationship between high

dietary cholesterol and CAD's risk; but, the minimal effect size of the risk was observed. A recent study [46], meta-analysis, and systemic reviews [47] do not conclude the dietary cholesterol as CAD's risk.

Several epidemiological studies reported that higher dietary fiber is associated with a reduced CAD risk [48]. Mechanistically, dietary fibers could lower atherosclerosis [49] and alter microbiota that modulates the immune system [50]. Besides, high fiber consumption is related to a higher intake of vitamins and minerals [51]. In divergence with the findings mentioned above, we could not observe that a higher fiber intake could lower CAD's risk. Moreover, the daily fiber intake amount was lower than the daily recommended allowance in both case and control groups, possibly because of specific dietary patterns in the Nepalese population. A prospective cohort study in Japan reported an inverse association of CHD with dietary intake of folic acid, vitamin B6, and vitamin B12 [18]. Recent dose-response meta-analysis shows that a higher intake of folate and vitamin B6 are associated with a lower risk of CAD [19]. Niacin intake has more enormous benefits in lowering LDL and increasing HDL in the blood in dyslipidemia patients; it could reduce the risk of CAD [52]. However, the Umbrella study concludes that nutritional supplements, such as folic acid, vitamin B6, vitamin B12 had no significant effect on mortality or CVDs outcomes [20]. In the present study, we did not observe any significant likelihood of CAD with dietary B vitamins. Vitamin A is associated with CAD severity, and β-carotene level diminish disease severity [21]. The present study revealed that vitamin C, vitamin A, and β-carotene intake were linked with CAD; a significant inverse association was reported with the intake of β-carotene but not with Vitamin A in the logistic model. This finding indicated antioxidant vitamins could be protective factors for the prevention of CAD in the Nepalese people. However, further large cohort or RCTs are required to confirm the present findings. A recent study in the Chinese population reported that β-carotene and vitamin C intake from the diet was inversely associated with deaths from all causes and CVDs in middle-aged or older people [53]. However, an inconsistent association of an antioxidant vitamin with CAD was reported in epidemiological studies [54].

Dietary calcium intake is inversely associated with CVDs [55]. Although a longer-term calcium intake is associated with a reduced risk of atherosclerosis, calcium supplementation may increase the risk of coronary artery calcification [56]. Dietary zinc intake was inversely associated with mortality from CHD in men but not women [57]. In contrast to these findings, our results did not reveal increased dietary calcium and zinc could reduce CAD's risk. In the present study, we observed a significant result of higher dietary iron intake associated with CAD's risk. Nonetheless, iron intake was not the topmost factors related to CAD in random forest analysis. The significant association of iron intake in the logistic model might be because of the multi-collinearity problem, which disguised the result. Simultaneously, heme iron intake was positively associated with CHD's risk in Western populations [58], where red meat is a primary dietary source of iron. In contrast, this relationship was negative in the Japanese, who receive heme iron from fish and shellfish [59]. Several studies have reported that CHD incidence was positively associated with ferritin levels and inversely associated with serum Fe and transferrin saturation [60].

In the present study, the data were only taken from non-fatal CAD, and the sample size (306 cases and 306 control)) was quite small. We excluded the patients with equal or more than 50% and less than 70% stenosis in any one main coronary artery branch. As TMT negative patients and patients with normal coronary angiography were included as a control group, they could have a possibility of microvascular angina. Moreover, the control group was selected from the same hospital with or without having cardiovascular risk factors; there was a possibility of selection bias in the study. Thus, generalization is the drawback of the present study. Moreover, since the study was a case-control study, it might have many inherent biases

that influence the causality link between the associated nutrients and the disease. Furthermore, our study observed dietary nutrients' association based on the food list commonly consumed by the Nepalese population. Another limitation was that we calculated only the values of eighteen nutrients available in Nepal food composition table 2017.

Despite these limitations, we tried to minimize the possible bias, such as recall bias, which was minimized by displaying food atlas for portion size and verifying the information with hospital data and patients' sick book as possible. Notably, we matched "Sex" individually and "Age" at the interval of five years to limit the confounders' effect. Moreover, patients who modified their dietary intake habit after being diagnosed as having any one of CAD's risk factors (obesity/hypertension/diabetes/dyslipidemia) were excluded to avoid possible unequal distribution of dietary nutrients due to predetermined health condition. Furthermore, the distribution of dietary nutrients in the control group was within the range of findings in a previous study conducted among Nepalese [61], which assured our control group represented the base/source population. Our study was unable to link the reverse causality between CAD and the dietary intake, which were likely altered as a result of a cardiovascular event; therefore, we used validated FFQ, which takes the information for recent one year, and also is considered elsewhere for research on nutrition concerning disease, so that we can compare the data with other analyses conveniently. We executed conditional logistic regression for model construction and random forest analysis to select nutrients variables, adjusting the collinearity problem to make the results valid.

## Conclusions

Diet plays a crucial role in the development of several non-communicable diseases, including CVDs. Resource-poor countries like Nepal, extracting the dietary nutrients of relative importance, could have rational and promising strategies for CAD prevention through dietary intervention. A combination of multiple factors rather than a single factor is more potent for nutrients intervention. Thus, a dietary intervention approach in CVDs is an effective strategy to reduce the public health burden. We conclude that dietary SFA, vitamin A RE, dietary total fat and oil, β-carotene, and cholesterol are the topmost five essential dietary nutrients associated with CAD development. Consistent with most of the studies, we report dietary SFA, total oil and fat, and cholesterol intakes are proportionately related, whereas β-carotene and vitamin C intakes are inversely related to CAD. Our study suggests a higher dietary intake of β-carotene and vitamin C are possible protective dietary nutrients, while an increased intake of dietary SFA, total fat and oil, and cholesterol are potential risk factors for CAD development. However, prospective cohort and RCTs studies with a large sample size are needed to explore the causal link of these nutrients for the risk of CAD development in the Nepalese population.

## Supporting information

**S1 Table. List of food items commonly consumed by Nepalese in Nepal.**
(DOCX)

**S2 Table. Mean daily nutrient intakes estimated by the average of three 24 h dietary recalls (DR) and two food frequency questionnaires (FFQs), and correlations between the two methods.** ICC: Intraclass correlation coefficient; [a]Data are shown as mean ± standard deviation (SD)
(DOCX)

**S3 Table. Distribution of nutritional factors associated with coronary artery disease between case and control groups in the study.** kcal: kilocalorie; g: gram; mg: milligram; mcg:

microgram; R.E.: retinol equivalent; PUFA: polyunsaturated fatty acid; MUFA: monounsaturated fatty acid; SFA: saturated fatty acid. [a]Paired t-test. [b]Mean and standard deviation (SD) value. $^*p \leq .05$; $^{**}p \leq .01$; $^{***}p \leq .001$.

(DOCX)

**S4 Table. Correlation matrix among eighteen dietary nutrients and energy intake.** PUFA: polyunsaturated fatty acid; MUFA: monounsaturated fatty acid; SFA: saturated fatty acid.

(DOCX)

**S1 File. Survey questionnaire set and informed consent in the Nepali language.**

(PDF)

**S2 File. Informed consent in the English language.**

(PDF)

**S3 File. Survey questionnaire set in the English language.**

(PDF)

**S4 File. Ethical approval letter from Nepal Health Research Council.**

(PDF)

**S5 File. Reporting checklist for case-control study based on STROBE guideline.**

(PDF)

**S6 File. Data set.**

(CSV)

**S7 File. R code for statistical analyses.**

(R)

## Acknowledgments

The authors are grateful to Shahid Gangalal National Heart Center in Nepal, where the hospital management assisted with the set-up and data collection. We also acknowledge Professor Aihua Gu and Dr. Xu Cheng (Nanjing Medical University) for directing the entire research work. The corresponding author had full access to all the data in the study and had final responsibility for the decision to submit for publication.

## Author Contributions

**Conceptualization:** Til Bahadur Basnet, Srijana G. C.

**Data curation:** Til Bahadur Basnet, Srijana G. C., Rajesh Basnet, Bidusha Neupane.

**Formal analysis:** Til Bahadur Basnet, Rajesh Basnet, Bidusha Neupane.

**Investigation:** Til Bahadur Basnet, Srijana G. C.

**Methodology:** Til Bahadur Basnet, Srijana G. C., Bidusha Neupane.

**Project administration:** Til Bahadur Basnet, Srijana G. C., Rajesh Basnet.

**Resources:** Til Bahadur Basnet, Srijana G. C., Rajesh Basnet.

**Software:** Til Bahadur Basnet, Bidusha Neupane.

**Supervision:** Til Bahadur Basnet.

**Validation:** Til Bahadur Basnet, Srijana G. C., Rajesh Basnet, Bidusha Neupane.

**Visualization:** Til Bahadur Basnet, Rajesh Basnet.

**Writing – original draft:** Til Bahadur Basnet.

**Writing – review & editing:** Til Bahadur Basnet, Srijana G. C., Rajesh Basnet, Bidusha Neupane.

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
