## [Decision Letter · Decision Letter 0]

27 May 2020

PONE-D-20-07225

Dietary nutrients of relative importance associated with coronary artery disease: Public health implication from random forest analysis

PLOS ONE

Dear Dr. BASNET,

Thank you for submitting your manuscript to PLOS ONE. After careful consideration, we feel that it has merit but does not fully meet PLOS ONE’s publication criteria as it currently stands. Therefore, we invite you to submit a revised version of the manuscript that addresses the points raised during the review process.

The manuscript lacks important methodological details needed for correct interpretation of the study. Please describe in better depth: (i) why your preferred to use “Random forest analysis”, and why both conditional LR and Random Forest Analysis were employed together to answer the same research question? (ii) Why these specific dietary nutrients were selected for analysis (why not others)? (iii) How did you select potential confounders for adjustment? (iv) How was the sample size of 306 cases and 306 controls determined? (v) what was the justification for developing two different multivariable models (model 1 and model 2) to explain an outcome while it is possible to have a single model with a better fit?It is not clear how some of the variables (e.g. physical exercise, smoking, alcohol use) were measured and categories. Can you please add a sub-section (under the methods section) that describes the variables of the study along with the measurement and classifications employed?Can you explain further how you detected very significant statistical difference in the age of the cases and controls after having exactly the same median and IQR values?It seems the authors have already published on risk factors of CHD using the same dataset before (https://pubmed.ncbi.nlm.nih.gov/31588344/). Please describe in the background section how this paper is different from the earlier and why it was not possible to report the findings of the two papers together.Most of the reported ORs only have borderline statistical significance. Based on these findings, would that be possible to given strong practical recommendations? Can you discuss the issue further?As noted by both of the reviewers, please discuss the limitations of study in a better depth. 

We look forward to receiving your revised manuscript.

Kind regards,

Samson Gebremedhin, PhD

Academic Editor

PLOS ONE

Other section-by-section comments

Abstract

Report separately the sample size for cases and controls. It would also be good to provide the concise enrolment criteria for cases and controls.

Background

This section is somehow superficial. Can you please add a paragraph that summarizes what is known about the dietary risk factors of CAD?

Methods and materials

Line 78-80: what about the 24-hr recall?Line 89-90: can you please describe the "24-h recall diet survey" in a better depth? In the results section, I don’t see any data on the results of this Criterion Validity assessment.Line 112: why did you use Wilcoxon sign ranked test for continuous variables? Was the distribution not normally distributed? Have you tried transformation before resorting to this non-parametric test?

Results

Line 129-31: “This might be because of the selection of the control groups from the outdoor patients of the same hospital where more hypertensive patients come for their heart checkup”. This should rather be presented in the “discussion” sectionTable 1: “Hard” > “vigorous”Table 1: drinking alcohol and smoking: have you tried consider dose of exposure while defining these variables?Line 134-143: is this from the 24-hr recall or the food frequency questionnaire?Table 2: please clearly denote significant differences with “*”Line 145-48: it is good that you have used conditional logistic regression. But what was the variable/s conditioned in the analysis. if “age” was the conditioning variable, then why was it adjusted in the model?

Journal Requirements:

3. Please amend the manuscript submission data (via Edit Submission) to include authors Ali Asghar Mirjat, Falak Zeb and Wiwik Indayati.

4. Your ethics statement must appear in the Methods section of your manuscript. If your ethics statement is written in any section besides the Methods, please move it to the Methods section and delete it from any other section. Please also ensure that your ethics statement is included in your manuscript, as the ethics section of your online submission will not be published alongside your manuscript.

Reviewers' comments:

Reviewer's Responses to Questions

**Comments to the Author**

1. Is the manuscript technically sound, and do the data support the conclusions?

Reviewer #1: Partly

Reviewer #2: No

2. Has the statistical analysis been performed appropriately and rigorously? 

Reviewer #1: No

Reviewer #2: No

3. Have the authors made all data underlying the findings in their manuscript fully available?

Reviewer #1: No

Reviewer #2: No

4. Is the manuscript presented in an intelligible fashion and written in standard English?

Reviewer #1: Yes

Reviewer #2: Yes

5. Review Comments to the Author

Reviewer #1: 1) Innovation perspective the research is not very significant for the researchers globally. Most of the innovations are already available in the literature. As far my knowledge many studies already published these kinds of results. For example, Mahalle et. al. (2016): Association of dietary factors with severity of coronary artery disease. Authors in the manuscript (line no. 257) state “Finally, we found a consistent result with other published studies.”

2) The primary outcome of any case-control study is mainly based on the quality data. If the data is not good quality data then the observations on these data don’t confirm any firm statement. Based on interview questionnaire and the response of the person don’t strongly establish the dietary quantity and quality. If the diet would be provided by any good supply source who would maintain the quantity and quality of the food consumed by the patients, then, the data quality would be much better.

3) There is no external validation of the results. It majorly focused on Nepalese dataset but the research title is generic. External validation (dataset collecting from near countries such as India, Bhutan, Bangladesh) is required for the title otherwise title need to be changed to focus on Nepalese population perspective.

4) The cohort is has been taken from one hospital and therefore, the results may be biased until the data has been taken in standard published process. At least 2 cohorts’ data is required for this study as it is based on face to face interview questions.

5) When we apply model’s then its strength needs to be checked by providing some metrics, such as, model’s stability by providing ROC curve, sensitivity, specificity, accuracy, f-score etc. These are clearly missing here.

6) There are several standard algorithms compared to Random Forest (RF) regression are available. This study chose RF without giving proper justification.

7) Case: control is used 1:1 which is pretty rare in real scenarios. This study has lacks in producing right case and control samples. For example, it doesn’t tell how many patients have participated in the study and how some of them have been excluded from the study with proper logic. By using only Angiography (<70% stenosis) is not enough.

8) In the discussion, some of the statements are contradict with the previous studies. But most of the cases, their rationalities are not explained clearly.

9) Discussion section is not well organized to understand the study as stated in the title.

10) No justification has been given on selecting confounding variables for adjustment.

11) For selecting confounding variables from one result to the next, there are several techniques like forward elimination, backward elimination could be used.

12) There are many statements stated without giving proper references, e.g., line no. 27, 29, 44, etc. and so on.

13) The full questionnaire and data are not exposed to assess the results.

14) Source code of the models and data are available to reproduce the results.

15) Some silly inconsistencies, like, line 144 states p<0.1.

Reviewer #2: Review

Manuscript Number: PONE-D-20-07225

Manuscript Title: Dietary nutrients of relative importance associated with coronary artery disease: Public health implication from random forest analysis

Summary of study

This is a retrospective case-control study examining the difference in calorie consumption and intake of 18 nutrients between those with established coronary artery disease (CAD) (n=306) versus a control (n=306). The purpose of the study is to establish associations between the consumption of specific nutrients and CAD in the Nepalese population; associations that the authors believe ought to inform future RCTs/prospective cohort studies and future health policy.

It examined a total of 612 patients from the Shahid Gangalal National Heart Centre, Nepal with case-control matched via age and gender. Patients were recruited to the study following admission to the hospital due to a suspected coronary problem (recent suspected myocardial infarction, an exercise induced positive stress test result, or who elected to have an angiography). Patients were assigned to the ‘case’ group if they showed evidence of severe stenosis (>70%). Control subjects were patients who upon arrival were found to have ‘normal’ angiographic results (authors should specify exactly what this means – the flow chart in Fig 1 gives the impression of 0% stenosis) or who were found to have no ECG abnormalities following an exercise stress test. According to the authors, those with intermediate stenosis (<70% obstruction) were excluded from the study, presumably any patient with evidence of stenosis ranging from 1–69%.

Data on nutrient consumption over the previous 12-months for each participant was derived from an EPIC food frequency questionnaire. Authors calculated average daily calorie consumption and selected 18 nutrients for analysis – total fat, total carbohydrate, total protein, fibre, calcium, phosphorous, iron, zinc, thiamine, riboflavin, niacin, Vitamin C, β-Carotene, Vitamin A, polyunsaturated fatty acids (PUFA), monounsaturated fatty acids (MUFA), saturated fatty acids (SFA), and cholesterol. Data on known risk factors (diabetes; dyslipidaemia; hypertension; obesity based on BMI; obesity based on central obesity via waist–hip ratio), and data on behaviours believed to be associated with CAD (alcohol consumption; smoking status; and physical activity level) were also recorded for each patient.

Using conditional multivariable logistic regression, authors report two statistically significant positive associations after controlling for known risk-factors (model-2): (i) a positive relationship with total fat intake (OR 1.13, 95% CI 1.05–1.21, P≤0.001); (ii) a positive relationship with dietary cholesterol intake (OR 1.06, 95% CI 1–1.12, P=0.02) – note, however, the inclusion of 1 in the confidence interval. The study also found statistically significant differences (inverse associations) between those with CAD and those without in terms of intakes of: (i) total carbohydrate intake (OR 0.93, 95% CI 0.86–0.99, P=0.04); (ii) calcium intake (OR 0.96, 95% CI 0.94–0.99, P=0.003); (iii) zinc intake (OR 0.88, 95% CI 0.79–0.98, P=0.02); (iv) niacin intake (OR 0.8, 95% CI 0.7–0.91, P≤0.001); (v) β-Carotene intake (OR 0.93; 95% CI 0.9–0.97, P=0.001).

Following this, the authors use a random forest regression to adjust for collinearity between the variables, which the authors believe allow them to discern the “five topmost important nutrients…linked with CAD: β-Carotene, fat, cholesterol, vitamin C, and fibre intakes”. However, the authors need to provide substantially more detail in the description of results and methodology used for this than is reported here. As it is, I’m not entirely sure what to make of these results.

They then proceed to offer a superficial comparison of their findings with the existing literature. This requires extensive revision, however, and the authors need to ensure they avoid committing citation bias here in regards to some of their claims. To do this, they need to show a greater understanding of the conflicting results of large RCTs, prospective cohort studies, and recent meta-analyses. Fortunately, for each of the nutrients examined here, there is an extensive literature on their relationship with CAD/CVD – so the authors really need to engage with this literature.

The authors conclude by claiming: “We conclude that dietary β-Carotene, total fat and oil, cholesterol, vitamin C, and fiber in the Nepalese population”.

However, I believe, the following points must be addressed before this study can be published:

1. Ethics

Before publication, the authors need to include more details about ethical approval. PLOS ONE’s policy on this is as follows:

"Human Subject Research (involving human participants and/or tissue)

- Give the name of the institutional review board or ethics committee that approved the study

- Include the approval number and/or a statement indicating approval of this research

- Indicate the form of consent obtained (written/oral) or the reason that consent was not obtained (e.g. the data were analysed anonymously))"

The authors state that this study was approved by the Nepal Health Research Council (NHRC). Is the value given here (308/2017-18) the study registration code? Is this associated with certification of ethical clearance by the NHRC ethics committee? Could the authors please attach the appropriate ethical approval documentation as a supplementary file; a letter from the NHRC stating this clearance was provided will suffice. As the authors state that they have received written informed consent by every study participant, could the authors also go into some detail about whether those participants were informed their data (anonymised) would be available open access?

2. Data availability

The authors must supply all data associated with their analyses. The authors have claimed all relevant data are supplied in the paper or supplementary files, but this is inaccurate. The results of this paper depend on the analysis of distribution data, and this is necessary for replication. PLOS ONE’s policy states – “PLOS journals require authors to make all data necessary to replicate their study’s findings publicly available without restriction at the time of publication”. Accordingly, the relevant data for every study participant on which this papers analysis depends must either be included as supplementary files or stored in an online data repository after patient data has been appropriately anonymised. These data could be provided in spreadsheets – for the 612 patients, this involves providing all data on control/case group membership, age, weight, gender, dyslipidaemia, hypertension, obesity, smoking, alcohol consumption, physical activity, calorie consumption, and all data regarding intake of the 18 examined nutrients. Indeed, in light of the following problems, it is the data collected by the authors here is probably the most important aspect of the study.

3. Problems concerning data analysis

Multiple comparisons:

One major problem is accounting for multiple comparisons in this study. For example, Table 2 lists 19 variables (18 nutrients and total calorie consumption) each of which has been compared with a Wilcoxon rank-sum test – so the quoted significance levels at the very least need to be adjusted for the fact that 19 comparisons have been made. The same issue recurs throughout the analysis. The authors should seek to resolve this issue via appropriate statistical methods. I would also recommend that this study is referred to a statistical editor upon these revisions to ensure that this issue has been appropriately resolved.

Multicollinearity:

Apart from the problem of multiple comparisons here there is also the problem that the data are not fully independent (multicollinearity) – specific micronutrients tend to be associated with other particular nutrients in different food types. Accordingly, it is important not to over interpret associations unless these issues are rigorously excluded. Accordingly, the authors should reflect more on this issue and adjust their methods/interpretations in light of this. One option here would be to extend their discussion of their random forest regression. Indeed, the paper would benefit from extending and deepening the description and results of this analysis, providing results on correlations between nutrient intakes and variance inflation factor. Again, I believe a statistical editor should be consulted.

Other data analysis issues:

In Table 1, the authors claim that the median age of authors is statistically significantly different between the case and control groups. The authors report the median age, interquartile range, and P-value as:

Case: 58 (50–65) | Control: 58 (50–65) | P=0.001

Not only is the median age the same, but so too are the interquartile ranges. Yet, despite this, the age difference is apparently statistically significant? The authors seem to interpret this result as meaningful:

“Because the age was matched in five year intervals in the study, the median age was 58 years, which was the same in both case and control groups, respectively, and still showing strong association (P=0.001).”

The authors claim that age was matched in 5-year intervals, but are we then to interpret these results as suggesting that there is actually a significant difference in age between case-control matched pairs? As CAD is strongly associated with age in previous research, this is important to clarify. To do this, the authors need to report the age distribution data for both groups as a supplementary file.

For the analysis in Table 2, the authors claim “as most of the study data were not normally distributed, median and interquartile range of nutrients are presented”. The authors should, therefore, include in their supplementary material the actual data underpinning their analysis. At the very least, they must include the mean, range, and SD for each nutrient, so the reader can understand exactly what the distribution of these data actually are. Indeed, the data provided in the paper is insufficient to replicate the necessary results reported, despite the authors declaration.

Statistical rhetoric:

The authors highlight a “highly significant” finding (p.9), this language is inappropriate and should be replaced. A result is simply either significant or non-significant and this is determined by whatever threshold of significance the authors deem necessary.

4. Problems of variable selection and measurement

Nutrient selection:

The authors select 18 nutrients to examine here, but why these specific nutrients are analysed is not adequately justified. Accordingly, the authors should make clearer why these items were selected for analysis.

In the supplementary file, a list of common foods consumed is provided. This raises further questions about why the authors chose only to analyse the variables they selected in this paper because other variables appear possible to derive from their data. For example, I see no reason why the amount of sugars in the diet couldn’t be calculated from the listed food items, so why isn’t this examined in the paper? Similarly, their decision to use total carbohydrate intake as a variable without breaking this down into refined and complex carbohydrates appears strange and problematic, particular because the authors acknowledge in the paper that there are important differences between these. Why then didn’t the authors calculate these?

In its current state, this Table of food items is both uninformative and misleading. It also raises further questions about how nutrient intakes were measured. Patients were asked about milk consumption, but the milk category does not clarify whether respondents were asked specifically about the amount of full, semi, or skimmed milk consumed, which would be necessary to understand fat content and fatty acid profiles, or whether this was a single category. Further questions about how the quantities of PUFA, MUFA, and SFA were calculated arise in regards to several of the vague categories, such as “vegetable oils”. Accordingly, the authors should include the specific dietary survey actually provided to patients. Furthermore, supplying the average amount of each food consumed by cases and controls for each item would shed more light on dietary habits. As recent research suggests different whole foods might have different effects on lipid profiles and thereby atherosclerosis, these data are important to report. At the very least the authors need to make available the intakes of each nutrient examined in this study per patient.

Other questions that arise are why were PUFA here considered as a single group and not split further into Omega-3 and Omega-6 variants? Why was the intake of trans-fats not measured? Thus, the authors need to revise the manuscript to give the reader a clearer understanding of the theoretical justification for the selection of the variables. As there is a voluminous literature on the relationship between diet and atherosclerosis/CHD/CVD extending back to the early 20th century, there is a wonderfully rich literature to draw from.

Self-reported nutritional data:

As all the nutritional data are all self-reported, the authors should include a clearly discussion in regards to their reliability given the known problems with this kind of data. I suspect there is a problem here. From Table 2, it appears that total daily nutritional intake was virtually the same in the two groups – despite the significantly higher incidence of obesity in the control group.

5. Referencing

In-text references in this paper appear occasionally only loosely related to the claims they are purported to be associated with. For example, reference number 2 is inserted after the following sentence:

“In Nepal, 30% of total death was related to cardiovascular disease (2).”

Yet, reference 2 is a paper by Rankinen et al. (2015), and nowhere in this paper is this claim made. Another reference chosen at random, reference 20, is used to support the authors claim that:

“Besides cholesterol is also an independent risk factor of CAD according to the lipid theory”

The paper referenced nowhere discussed dietary cholesterol. It is a paper examining, as the title suggests, the “Relationships Between Components of Blood Pressure and Cardiovascular Events in Patients with Stable Coronary Artery Disease and Hypertension”. The only mention of cholesterol in this paper is HDL-C and LDL-C – that is, cholesterol bound in particular classes of lipoproteins carried in the blood.

If the authors make the rest of the revisions outlined, I will examine each of the references of this paper. So my recommendation would be to go through each reference and ensure it is relevant to the claim being made. As discussed, the authors also need to ensure they have adequately represented the state of research in relation to their claims. If the article is resubmitted, I’ll check each

6. Flawed study design

However, there is one problem that may undermine the point of revising this manuscript. The authors have a case-matched control group – but the control group are not healthy individuals, but patients with other health conditions and cardiovascular symptoms. This is clearly evident by the way the authors chose to enrol patients – all patients were being examined because of suspected coronary problems. This makes it impossible to talk of differences between these groups in terms of risk factors.

For example, looking at Table 1, the ‘control’ group has a significantly higher incidence of obesity and central obesity – but it would be obviously wrong to conclude that obesity and central obesity are protective against cardiovascular disease. Here we’re seeing a stratification of phenotypic characteristics between two different patient groups, and from this we can’t conclude anything at all about risk.

This might also explain the extremely strange finding that the number of hypertensives was roughly the same in both the control and case groups - Control: 143 hypertensives (~46.7%) | Case: 142 hypertensives (~46.4%). As hypertension is one of the key known risk factors in the development of CAD/CVD and extensively supported in the literature, this finding requires a lot more reflection. Why were hypertensives so common in the control group? This control group had apparently no evidence of stenosis – so this seems to be quite an important avenue to explore what went on here.

Accordingly, this design is inappropriate for the authors stated intention: “The present case-control study was designed to determine the association of dietary nutrients with CAD in the Nepalese population”.

If this study is to be published, the authors need to somehow explain why this control group can be considered representative of a broader population. Later in the paper the authors do highlight the results may have been biased due to “the selection of the control group from the outdoor patients from the same hospital where more hypertensive patients come for their heart check-up”, but this seems to critically undermine the entire results of this study.

6. PLOS authors have the option to publish the peer review history of their article (what does this mean?). If published, this will include your full peer review and any attached files.

Reviewer #1: No

Reviewer #2: No

---

## [Author Response · Author response to Decision Letter 0]

16 Aug 2020

Response Letter

July 30, 2020

To,

Dear Academic editors and Reviewers,

We highly appreciate comments of academic editor and both the reviewers in our manuscript entitled " Dietary nutrients of relative importance associated with coronary artery disease: Public health implication from random forest analysis" (PONE-D-20-07225). We have modified the manuscript; accordingly, there has been a visible improvement in our manuscript after incorporating the comments. The responses to the comments are listed below.

We hope this manuscript will be acceptable for publication in your reputed journal.

Sincerely,

Til Bahadur Basnet

Ph.D. Candidate (Epidemiology and health statistics)

………………………………………..

1. Academic editor comments

• The manuscript lacks important methodological details needed for correct interpretation of the study. Please describe in better depth: (i) why your preferred to use “Random forest analysis”, and why both conditional LR and Random Forest Analysis were employed together to answer the same research question? (ii) Why these specific dietary nutrients were selected for analysis (why not others)? (iii) How did you select potential confounders for adjustment? (iv) How was the sample size of 306 cases and 306 controls determined? (v) what was the justification for developing two different multivariable models (model 1 and model 2) to explain an outcome while it is possible to have a single model with a better fit?

Response: Thank you so much for all the comments, which were very valuable and encouraging. We believe that we have improved the methodological details for correct interpretation of the study addressing the quarries raised. 

(i) why your preferred to use “Random forest analysis”, and why both conditional LR and Random Forest Analysis were employed together to answer the same research question?

Random forest (RF) analysis is an algorithm based technique that not only picks up the important variables but also useful technique when the data have normality as well as multicollinearity problem. RF is a useful technique when data have relatively fewer observations because the analysis is based on an ensemble of classification trees in which it split the data into several nodes that maximized the homogeneity in each group, the random forest assembled hundreds more classification trees with a selection of correlates randomly. However, it cannot determine the strength of association. In contrast, conditional LR is the correct method to determine the strength of association in matched case control study. Therefore, we preferred both analyses in our study.

(ii) Why these specific dietary nutrients were selected for analysis (why not others)?

Upon review of the literature, dietary nutrients that we selected were associated with coronary artery disease (CAD) with the discrepancy in results in different populations across the globe. And we calculated nutrient intake based on "Nepal food composition table 2017," which lacks other specific nutrients like biotin, magnesium, selenium, and others related to CAD.

(iii) How did you select potential confounders for adjustment?

Although there are several newer techniques like a-priori change (e.g., 10% or 15%) in the effect estimate criteria, bias-variance tradeoff, directed acyclic graph (DAG), we adopted conventional way of a p-value of cut off in the statistical model. After bivariate analysis (Wilcoxon sign-rank sum test for continuous but non-parametric paired data and McNemar test for dichotomous paired data analysis in matched case-control study ) [1], those variables which showed p-value less than 0.1 ( usually ranges from 0.05 up to 0.2) in bivariate analysis has been considered as potential confounders for adjustments. 

(iv) How was the sample size of 306 cases and 306 controls determined?

The sample size was calculated for matched case-control study with group of unequal sample size taking Zα=0.05, Zβ=0.8, prevalence of exposure = 0.19 (prevalence of smoking in the Nepalese population [2]), minimum detectable OR = 1.75. Using the formula developed by Schlesselman, the minimum sample size was 279. After adding a 10% non-response rate, the sample size was 306 in each group.

However, the present work was to construct a model using conditional logistic regression analysis. For multivariable analysis, the number of observations, in general, is ten times the number of predictors was suggested [3]. In this study, 19 nutrients variables and ten socio-demographic and cardio-metabolic risk factors were considered. Indeed, we selected only 23 predictors in the initial step of model construction. In the final model, only 12 variables were considered significantly associated with the outcome variables. Thus, we believe our sample size was considered enough for the present research work. 

(v) What was the justification for developing two different multivariable models (model 1 and model 2) to explain an outcome while it is possible to have a single model with a better fit?

Thank you for your suggestion to develop a single model with a better fit. We have constructed a single energy-adjusted parsimonious multivariate model with a better fit by stepwise backward deletion process. And the results from the final model were presented in table 4.

• It is not clear how some of the variables (e.g. physical exercise, smoking, alcohol use) were measured and categories. Can you please add a sub-section (under the methods section) that describes the variables of the study along with the measurement and classifications employed?

We acknowledge your suggestions. So, we included the operational definition of conventional variables (e.g., physical exercise, smoking, alcohol use) as a sub-section under the methods section and described how they were measured and categorized.

• Can you explain further how you detected very significant statistical difference in the age of the cases and controls after having exactly the same median and IQR values?

Our study was matched case-control study, and we exactly matched the “sex," but we matched the “age” at an interval of 5 years. Because of interval matching and Wilcoxon-sign ranked test for matched paired non-parametric data, “age” showed a significant association with outcome. 

• It seems the authors have already published on risk factors of CHD using the same dataset before (https://pubmed.ncbi.nlm.nih.gov/31588344/). Please describe in the background section how this paper is different from the earlier and why it was not possible to report the findings of the two papers together.

The paper which we published was the “association of smoking with CAD." If we included the dietary nutrients in that article, that would be lager in word count, which was beyond an essential requirement for the journal. Also, the paper would not be specific and exciting to the reader. Therefore, because of the words limit in the journal and specific detailed description of factors associated, it was not possible to include the dietary factors in the previous manuscript.

• Most of the reported ORs only have borderline statistical significance. Based on these findings, would that be possible to given strong practical recommendations? Can you discuss the issue further?

Unlike conventional risk factors (such as hypertension, diabetes, smoking), there was still a discrepancy in findings of the association of dietary nutrients with CAD in different populations across the world. Furthermore, several studies have reported relatively a higher effect size for conventional risk factors and smaller effect size for nutritional risk factors. We also observed some essential nutrients (beta-carotene, riboflavin, fiber, and vitamin C) that have not met the minimum recommended daily requirements (RDA) both in cases and controls. To confirm the effect of nutrients specific such as antioxidant vitamins on CAD, RCT and large cohort studies are required in the Nepalese population.

• As noted by both of the reviewers, please discuss the limitations of study in a better depth. 

Thank you much for scrupulous perusing and valuable comments for making quality paper publishable in PLoS One, a world-famous journal. We included all the feedback provided by both reviewers, including the limitations of the study noted by reviewers. 

Other section-by-section comments

Abstract

Report separately the sample size for cases and controls. It would also be good to provide the concise enrolment criteria for cases and controls.

Now, we included the sample size for cases and control in the abstract. Also, we have written the concise enrolment criteria for cases and controls.

Background

• This section is somehow superficial. Can you please add a paragraph that summarizes what is known about the dietary risk factors of CAD?

As per your suggestion, we have included one paragraph briefly describing the dietary risk factors of CAD. 

Methods and materials

• Line 78-80: what about the 24-hr recall?

• Line 89-90: can you please describe the "24-h recall diet survey" in a better depth? In the results section, I don’t see any data on the results of this Criterion Validity assessment.

• Line 112: why did you use Wilcoxon sign ranked test for continuous variables? Was the distribution not normally distributed? Have you tried transformation before resorting to this non-parametric test?

We appreciate the comments that assisted a lot in improving our manuscript.

• 24-hr recall has been described in the manuscript. 

• Before collecting the information from 306 cases and 306 controls, we carried out a survey among 115 OPD patients in the hospital to determine the validity and reproducibility of FFQ customized to Nepalese food items. Now the result has been incorporated as a supplementary file.

• Because the data were not normally distributed, we determined to apply a non-parametric test. We believe an interpretation of transformed data misleads our result in bivariate analysis. Wilcoxon sign-rank sum test for continuous but non-parametric paired data and McNemar test for dichotomous paired data are the correct tests for the analyses in matched case-control study [1]. 

Results

• Line 129-31: “This might be because of the selection of the control groups from the outdoor patients of the same hospital where more hypertensive patients come for their heart checkup”. This should rather be presented in the “discussion” section

• Table 1: “Hard” > “vigorous”

• Table 1: drinking alcohol and smoking: have you tried consider dose of exposure while defining these variables?

• Line 134-143: is this from the 24-hr recall or the food frequency questionnaire?

• Table 2: please clearly denote significant differences with “*”

• Line 145-48: it is good that you have used conditional logistic regression. But what was the variable/s conditioned in the analysis. if “age” was the conditioning variable, then why was it adjusted in the model?

Line 129-31: We appreciate the comments and have written it in the discussion section.

• We have changed "hard" into "vigorous."

• Our questionnaire was designed to collect the dose of exposure to alcohol use and smoking. However, because of inconsistency in responses, we resorted to categorical variables.

• The information mentioned in the Line 134-143 was from the food frequency questionnaire. We now described the source of information in the text.

• In Table 2, we now clearly denoted significant differences with “*”.

• Line 145-48: Matching in a case-control study does not control for confounding by the matching factors. A matched design may require controlling for the matching factors in the analysis [4]. In our study, we exactly matched the "sex," but we matched the "age" at an interval of 5 years. Because of interval matching and Wilcoxon-sign ranked test for paired data, “age" showed a significant association with outcome. Therefore, even though "sex" and "age" were conditioning variables in the study, we adjusted “age” in the model. However, it had been removed in step-wise backward deletion process while developing the single parsimonious model as per your suggestion. 

Journal Requirements:

 We have prepared the manuscript according to PLOS ONE's style requirements, including those for file naming.

We are submitting the survey questionnaire in both the original language and English, as Supporting information. 

3. Please amend the manuscript submission data (via Edit Submission) to include authors Ali Asghar Mirjat, Falak Zeb and Wiwik Indayati.

Now, we amended the manuscript submission data.

4. Your ethics statement must appear in the Methods section of your manuscript. If your ethics statement is written in any section besides the Methods, please move it to the Methods section and delete it from any other section. Please also ensure that your ethics statement is included in your manuscript, as the ethics section of your online submission will not be published alongside your manuscript.

We appreciated the advice to include the ethical statement. We have written it in the Method section of our manuscript.

 We included the captions for supporting information files at the end of the manuscript and updated in-text citations. 

Reviewer #1: 

1) Innovation perspective the research is not very significant for the researchers globally. Most of the innovations are already available in the literature. As far my knowledge many studies already published these kinds of results. For example, Mahalle et. al. (2016): Association of dietary factors with severity of coronary artery disease. Authors in the manuscript (line no. 257) state “Finally, we found a consistent result with other published studies.”

Thank you so much for the comments. This matched case-control study was carried out to determine the association of dietary nutrients with CAD among Nepalese was the first case-control study. We believe that differences in the dietary habit of the different populations across space and time warrant a broader understanding of the dietary nutrients related to a specific disease. 

2) The primary outcome of any case-control study is mainly based on the quality data. If the data is not good quality data then the observations on these data don’t confirm any firm statement. Based on interview questionnaire and the response of the person don’t strongly establish the dietary quantity and quality. If the diet would be provided by any good supply source who would maintain the quantity and quality of the food consumed by the patients, then, the data quality would be much better.

We acknowledge that the quality data are the most for the interpretation of the case-control study. Although a 24h dietary recall survey is considered a standard dietary survey tool, it can only collect the prier 24h diet information. Therefore, we used customized validated FFQ to collect the diet information during the past 12 months, and the average was calculated. Thus, we believe short term supply of quality food in the hospital or home would not have changed our entire data.

3) There is no external validation of the results. It majorly focused on Nepalese dataset but the research title is generic. External validation (dataset collecting from near countries such as India, Bhutan, Bangladesh) is required for the title otherwise title need to be changed to focus on Nepalese population perspective.

We appreciate for your suggestion. We have mentioned the study population in the methods section and external validation in the limitation of the study section.

4) The cohort is has been taken from one hospital and therefore, the results may be biased until the data has been taken in standard published process. At least 2 cohorts’ data is required for this study as it is based on face to face interview questions.

In Nepal, there are only two public hospitals, namely Shahid Gangalal National Heart Center and Monmon Cardiothoracic Treatment Center, which are specially equipped to treat heart-related health problems. We took permission from both hospitals and got ethical approval from the national health research council in Nepal. However, we reported very few cases (~5 cases in a month) from Monmon Cardiothoracic Treatment Center. We determined to carry out in Shahid Gangalal National Heart Center, a national level cardiac referral center in the country. 

5) When we apply model’s then its strength needs to be checked by providing some metrics, such as, model’s stability by providing ROC curve, sensitivity, specificity, accuracy, f-score etc. These are clearly missing here.

Thank you for your suggestion. We have included the ROC curve and a comparison table about sensitivity, specificity, accuracy, and f-score of the two regression analysis models.

6) There are several standard algorithms compared to Random Forest (RF) regression are available. This study chose RF without giving proper justification.

We used random forest analysis because our data had a multicollinearity problem. Also, RF analysis is famous in predicting diseases in the medical field [5-7]. In the revised manuscript, we have provided the proper justification in statistical analyses of the "method" section.

7) Case: control is used 1:1 which is pretty rare in real scenarios. This study has lacks in producing right case and control samples. For example, it doesn’t tell how many patients have participated in the study and how some of them have been excluded from the study with proper logic. By using only Angiography (<70% stenosis) is not enough.

Although we have not reported the number of a non-participant in the study, we had collected the dietary information according to our predetermined inclusion and exclusion criteria that we mentioned in the "method" section. 

8) In the discussion, some of the statements are contradict with the previous studies. But most of the cases, their rationalities are not explained clearly.

We believe that the discussion section has modified to contrast and support our study findings based on available literature. 

9) Discussion section is not well organized to understand the study as stated in the title.

Our title is the relative importance of dietary nutrients related to CAD. We believe that we have discussed all nutrients associated with CAD in the study.

10) No justification has been given on selecting confounding variables for adjustment.

Although there are several newer techniques like a-priori change (e.g., 10% or 15%) in the effect estimate criteria, bias-variance tradeoff, directed acyclic graph, etc., we adopted conventional way of a p-value of cut off in the statistical model. After bivariate analysis (Wilcoxon sign-rank sum test for continuous but non-parametric paired data and McNemar test for dichotomous paired data analysis in matched case-control study ) [1], those variables which showed p-value less than 0.1 ( usually ranges from 0.05 up to 0.2) in bivariate analysis has been considered as potential confounders for adjustments. 

11) For selecting confounding variables from one result to the next, there are several techniques like forward elimination, backward elimination could be used.

We used step-wise backward elimination technique while selecting confounding variables from one result to next.

12) There are many statements stated without giving proper references, e.g., line no. 27, 29, 44, etc. and so on.

We added the citation in the statements (line no. 27, 29, 44,) and also where references were missing.

13) The full questionnaire and data are not exposed to assess the results.

We have now submitted the full questionnaire and data as a supplementary file.

14) Source code of the models and data are available to reproduce the results.

We now have submitted the source code of R of the model and data as a supplementary file.

15) Some silly inconsistencies, like, line 144 states p<0.1.

While considering potential confounding factors, we have read that some epidemiological papers were taking variables not only p-value up to <0.2 but also the significant cofounding variables in other literature in the model development process. However, we now determined to resort to 0.05 as per your suggestion.

#Reviewer 2

1. Ethics

Before publication, the authors need to include more details about ethical approval. PLOS ONE’s policy on this is as follows:

"Human Subject Research (involving human participants and/or tissue)

- Give the name of the institutional review board or ethics committee that approved the study

- Include the approval number and/or a statement indicating approval of this research

- Indicate the form of consent obtained (written/oral) or the reason that consent was not obtained (e.g. the data were analysed anonymously))"

The authors state that this study was approved by the Nepal Health Research Council (NHRC). Is the value given here (308/2017-18) the study registration code? Is this associated with certification of ethical clearance by the NHRC ethics committee? Could the authors please attach the appropriate ethical approval documentation as a supplementary file; a letter from the NHRC stating this clearance was provided will suffice. As the authors state that they have received written informed consent by every study participant, could the authors also go into some detail about whether those participants were informed their data (anonymised) would be available open access?

Yes, the value given (308/2017-18) is the study registration code. Of course, this is associated with the certification of ethical clearance by the Nepal Health Research Council (NHRC) ethics committee. According to your suggestion, we are pleased to attach the ethical approval documentation as a supplementary file: a letter from the NHRC stating this clearance. However, we did not inform participants into some detail about their data (anonymised) would be available open access.

2. Data availability

The authors must supply all data associated with their analyses. The authors have claimed all relevant data are supplied in the paper or supplementary files, but this is inaccurate. The results of this paper depend on the analysis of distribution data, and this is necessary for replication. PLOS ONE’s policy states – “PLOS journals require authors to make all data necessary to replicate their study’s findings publicly available without restriction at the time of publication”. Accordingly, the relevant data for every study participant on which this papers analysis depends must either be included as supplementary files or stored in an online data repository after patient data has been appropriately anonymised. These data could be provided in spreadsheets – for the 612 patients, this involves providing all data on control/case group membership, age, weight, gender, dyslipidaemia, hypertension, obesity, smoking, alcohol consumption, physical activity, calorie consumption, and all data regarding intake of the 18 examined nutrients. Indeed, in light of the following problems, it is the data collected by the authors here is probably the most important aspect of the study.

Thank you for your suggestion. We have now submitted all the data in spreadsheets on control/case group membership, including all variables under study.

3. Problems concerning data analysis

Multiple comparisons:

One major problem is accounting for multiple comparisons in this study. For example, Table 2 lists 19 variables (18 nutrients and total calorie consumption) each of which has been compared with a Wilcoxon rank-sum test – so the quoted significance levels at the very least need to be adjusted for the fact that 19 comparisons have been made. The same issue recurs throughout the analysis. The authors should seek to resolve this issue via appropriate statistical methods. I would also recommend that this study is referred to a statistical editor upon these revisions to ensure that this issue has been appropriately resolved.

Our study was about variable selection and model construction. It was not about evaluating the association of each variable and the outcome, and nineteen nutrients were not the categories of a single variable. Each variable was separately tested with a Wilcoxon sign rank test for the selection of potential variables for model construction. Thus, we believe that there were not multiple comparison problems in our data set.

Multicollinearity:

Apart from the problem of multiple comparisons here there is also the problem that the data are not fully independent (multicollinearity) – specific micronutrients tend to be associated with other particular nutrients in different food types. Accordingly, it is important not to over interpret associations unless these issues are rigorously excluded. Accordingly, the authors should reflect more on this issue and adjust their methods/interpretations in light of this. One option here would be to extend their discussion of their random forest regression. Indeed, the paper would benefit from extending and deepening the description and results of this analysis, providing results on correlations between nutrient intakes and variance inflation factor. Again, I believe a statistical editor should be consulted.

Thank you for your valuable suggestion for improving the quality of the manuscript. As per academic editor recommendation, we determined to construct a single model by executing the conditional logistic regression. While making a single parsimony model, we excluded one of the variables that showed a multicollinearity problem deciding based on the square root of vif value >2 as a cutup point. However, excluding one of the variables undermines the importance of that for the risk of disease. Therefore, we performed random forest analysis, including all variables in the model, to get the most important variables associated with CAD. 

Other data analysis issues:

In Table 1, the authors claim that the median age of authors is statistically significantly different between the case and control groups. The authors report the median age, interquartile range, and P-value as:

Case: 58 (50–65) | Control: 58 (50–65) | P=0.001

Not only is the median age the same, but so too are the interquartile ranges. Yet, despite this, the age difference is apparently statistically significant? The authors seem to interpret this result as meaningful:

“Because the age was matched in five year intervals in the study, the median age was 58 years, which was the same in both case and control groups, respectively, and still showing strong association (P=0.001).”

The authors claim that age was matched in 5-year intervals, but are we then to interpret these results as suggesting that there is actually a significant difference in age between case-control matched pairs? As CAD is strongly associated with age in previous research, this is important to clarify. To do this, the authors need to report the age distribution data for both groups as a supplementary file.

Our study was a matched case-control study, and we exactly matched the “sex," but we matched the “age” at an interval of 5 years. Because of interval matching and Wilcoxon-sign ranked test for paired data, “age” was showing significant association with outcome. Wilcoxon sign-rank sum test for non-parametric paired data and McNemar test for dichotomous paired data are the correct test for the analysis in the matched case-control study [1]. Matching in a case-control study does not control for confounding by the matching factors. A matched design may require controlling for the matching factors in the analysis [4]. Therefore, even though “sex” and “age” were conditioning variables in the study, we adjusted “age” in the model. 

For the analysis in Table 2, the authors claim “as most of the study data were not normally distributed, median and interquartile range of nutrients are presented”. The authors should, therefore, include in their supplementary material the actual data underpinning their analysis. At the very least, they must include the mean, range, and SD for each nutrient, so the reader can understand exactly what the distribution of these data actually are. Indeed, the data provided in the paper is insufficient to replicate the necessary results reported, despite the authors declaration.

We followed the standard statistical norms for the presentation of data. We believe that it looks awkward, putting two averages (median and mean) in a single table representing the average of the same variables. As per your suggestion, we have presented the data in the supplementary table that includes mean, range, and SD (S3 Table). 

Statistical rhetoric:

The authors highlight a “highly significant” finding (p.9), this language is inappropriate and should be replaced. A result is simply either significant or non-significant and this is determined by whatever threshold of significance the authors deem necessary.

We appreciate your suggestion. We just changed the language as per your suggestion that “highly significant” to only either significant or non-significant.

4. Problems of variable selection and measurement

Nutrient selection:

The authors select 18 nutrients to examine here, but why these specific nutrients are analysed is not adequately justified. Accordingly, the authors should make clearer why these items were selected for analysis.

We calculated daily nutrient intake from Nepalese, commonly consumed food items based on "Nepal food composition table 2017," which lacks the values of all nutrients per 100 gm food items are available for the food items. Although we found literature where these 18 nutrients were mostly related to CAD, the results were inconsistent in the different populations across the world.

In the supplementary file, a list of common foods consumed is provided. This raises further questions about why the authors chose only to analyse the variables they selected in this paper because other variables appear possible to derive from their data. For example, I see no reason why the amount of sugars in the diet couldn’t be calculated from the listed food items, so why isn’t this examined in the paper? Similarly, their decision to use total carbohydrate intake as a variable without breaking this down into refined and complex carbohydrates appears strange and problematic, particular because the authors acknowledge in the paper that there are important differences between these. Why then didn’t the authors calculate these?

Yes, we acknowledged that we did not breakdown the total carbohydrate into "refined" and "complex." We have now mentioned this issue in the limitation section.

In its current state, this Table of food items is both uninformative and misleading. It also raises further questions about how nutrient intakes were measured. Patients were asked about milk consumption, but the milk category does not clarify whether respondents were asked specifically about the amount of full, semi, or skimmed milk consumed, which would be necessary to understand fat content and fatty acid profiles, or whether this was a single category. Further questions about how the quantities of PUFA, MUFA, and SFA were calculated arise in regards to several of the vague categories, such as “vegetable oils”. Accordingly, the authors should include the specific dietary survey actually provided to patients. Furthermore, supplying the average amount of each food consumed by cases and controls for each item would shed more light on dietary habits. As recent research suggests different whole foods might have different effects on lipid profiles and thereby atherosclerosis, these data are important to report. At the very least the authors need to make available the intakes of each nutrient examined in this study per patient.

Regarding milk category, Nepalese in rural area usually have a practice of making butter: heating milk and allow cooling down and extracting the butter before consumption of milk in a village. In city area, the marketable milk is semi skimmed milk. Therefore, we calculated categories of fat based on semi skimmed milk. Regarding vegetable oil, we included nutrients value separately (soybean oil, mustard oil, sunflower oil). We did not calculate effect of the different whole food on the lipid profile of patients.

Other questions that arise are why were PUFA here considered as a single group and not split further into Omega-3 and Omega-6 variants? Why was the intake of trans-fats not measured? Thus, the authors need to revise the manuscript to give the reader a clearer understanding of the theoretical justification for the selection of the variables. As there is a voluminous literature on the relationship between diet and atherosclerosis/CHD/CVD extending back to the early 20th century, there is a wonderfully rich literature to draw from.

We calculated nutrients based on the "Nepal food composition table 2017," which lacks all nutrients value for the given food items. This limitation, we have now mentioned in the limitation of the study. 

Self-reported nutritional data:

As all the nutritional data are all self-reported, the authors should include a clearly discussion in regards to their reliability given the known problems with this kind of data. I suspect there is a problem here. From Table 2, it appears that total daily nutritional intake was virtually the same in the two groups – despite the significantly higher incidence of obesity in the control group.

According to your suggestion, we have discussed the problem related to the self-reported diet survey. We did not report a higher incidence of obesity in the control group. I think the word “Normal” was making confusion. Therefore, we change the word “Normal” to "No." Now, we reported the BMI for Asian ≥27.5 kg/m2 as obesity. So, now the figures in the Table are different from the previous values.

5. Referencing

In-text references in this paper appear occasionally only loosely related to the claims they are purported to be associated with. For example, reference number 2 is inserted after the following sentence:

“In Nepal, 30% of total death was related to cardiovascular disease (2).”

Yet, reference 2 is a paper by Rankinen et al. (2015), and nowhere in this paper is this claim made. Another reference chosen at random, reference 20, is used to support the authors claim that:

“Besides cholesterol is also an independent risk factor of CAD according to the lipid theory”

The paper referenced nowhere discussed dietary cholesterol. It is a paper examining, as the title suggests, the “Relationships Between Components of Blood Pressure and Cardiovascular Events in Patients with Stable Coronary Artery Disease and Hypertension”. The only mention of cholesterol in this paper is HDL-C and LDL-C – that is, cholesterol bound in particular classes of lipoproteins carried in the blood.

If the authors make the rest of the revisions outlined, I will examine each of the references of this paper. So my recommendation would be to go through each reference and ensure it is relevant to the claim being made. As discussed, the authors also need to ensure they have adequately represented the state of research in relation to their claims. If the article is resubmitted, I'll check each.

We are very regretful to inform that while converting from other reference styles (Vancouver) to PLoS one, all reference lists had been mismatched because of some technical problem. Now, we corrected all citation-related issues in the manuscript.

6. Flawed study design

However, there is one problem that may undermine the point of revising this manuscript. The authors have a case-matched control group – but the control group are not healthy individuals, but patients with other health conditions and cardiovascular symptoms. This is clearly evident by the way the authors chose to enrol patients – all patients were being examined because of suspected coronary problems. This makes it impossible to talk of differences between these groups in terms of risk factors.

For example, looking at Table 1, the ‘control’ group has a significantly higher incidence of obesity and central obesity – but it would be obviously wrong to conclude that obesity and central obesity are protective against cardiovascular disease. Here we’re seeing a stratification of phenotypic characteristics between two different patient groups, and from this we can’t conclude anything at all about risk. 

In table 1, we did not report a higher prevalence of obesity and central obesity. The percentage we mentioned were about “normal” and also provided in the foot notes. Now, we realized that the word “Normal” is not appropriate and unclear. Therefore, we changed “Normal” to “No”.

This might also explain the extremely strange finding that the number of hypertensives was roughly the same in both the control and case groups - Control: 143 hypertensives (~46.7%) | Case: 142 hypertensives (~46.4%). As hypertension is one of the key known risk factors in the development of CAD/CVD and extensively supported in the literature, this finding requires a lot more reflection. Why were hypertensives so common in the control group? This control group had apparently no evidence of stenosis – so this seems to be quite an important avenue to explore what went on here.

Although hypertension is one of the critical risk factors in the development of CAD/CVD, the same prevalence was reported because the selection of the control groups was from the outdoor patients of the same hospital where more hypertensive patients had come for their hearts checkup. However, we aimed to determine the nutrients factors, not cardiometabolic factors related to CAD. We believe more the matching of the demographic and cardiometabolic factors more precisely can detect the effect size of nutrients to CAD.

Our control group was those patients who were having either stress test negative or normal angiography. Those patients who were recently diagnosed as hypertension apparently did not show stenosis. Mostly the diagnosed patients were taking medicine to control hypertension and dyslipidemia. Notably, we classified them as hypertensive patients.

Accordingly, this design is inappropriate for the authors stated intention: “The present case-control study was designed to determine the association of dietary nutrients with CAD in the Nepalese population”.

If this study is to be published, the authors need to somehow explain why this control group can be considered representative of a broader population. Later in the paper the authors do highlight the results may have been biased due to “the selection of the control group from the outdoor patients from the same hospital where more hypertensive patients come for their heart check-up”, but this seems to critically undermine the entire results of this study.

Our study was focusing on the dietary nutrients related to CAD. Principally, in the case-control study, the matching of the confounding variables, better can predict the risk posed by the intended risk factors (in our study dietary nutrients). We also believe that almost equal distribution of metabolic risk factors in both cases and controls, the mediation effect posed by these metabolic risk factors (obesity, hypertension, and dyslipidemia) were adjusted.

 Thank you very much for the critical appraisal of our manuscript, which has helped improve the manuscript quality, making it publishable in a highly famous journal "PLoS One." 

 References

1. Conway A, Rolley JX, Fulbrook P, Page K, Thompson DR. Improving statistical analysis of matched case-control studies. Research in nursing & health. 2013;36(3):320-4. Epub 2013/02/15. doi: 10.1002/nur.21536. PubMed PMID: 23408517.

2. Aryal K, Neupane S, Mehata S, Vaidya A, Singh S, Paulin F, et al. Non Communicable Diseases Risk Factors: STEPS Survey Nepal 20132014.

3. Altman DG. Practical statistics for medical research. London: Chapman & Hall, 1991.

4. Pearce N. Analysis of matched case-control studies. BMJ (Clinical research ed). 2016;352:i969. Epub 2016/02/27. doi: 10.1136/bmj.i969. PubMed PMID: 26916049; PubMed Central PMCID: PMCPMC4770817 interests and declare the following: none. Provenance and peer review: Not commissioned; externally peer reviewed.

5. Beunza JJ, Puertas E, García-Ovejero E, Villalba G, Condes E, Koleva G, et al. Comparison of machine learning algorithms for clinical event prediction (risk of coronary heart disease). Journal of biomedical informatics. 2019;97:103257. Epub 2019/08/03. doi: 10.1016/j.jbi.2019.103257. PubMed PMID: 31374261.

6. Brisimi TS, Xu T, Wang T, Dai W, Adams WG, Paschalidis IC. Predicting Chronic Disease Hospitalizations from Electronic Health Records: An Interpretable Classification Approach. Proceedings of the IEEE Institute of Electrical and Electronics Engineers. 2018;106(4):690-707. Epub 2019/03/20. doi: 10.1109/jproc.2017.2789319. PubMed PMID: 30886441; PubMed Central PMCID: PMCPmc6419763.

7. Zhang X, Dai Z, Lau EHY, Cui C, Lin H, Qi J, et al. Prevalence of bone mineral density loss and potential risk factors for osteopenia and osteoporosis in rheumatic patients in China: logistic regression and random forest analysis. Annals of translational medicine. 2020;8(5):226. Epub 2020/04/21. doi: 10.21037/atm.2020.01.08. PubMed PMID: 32309373; PubMed Central PMCID: PMCPmc7154412.

---

## [Decision Letter · Decision Letter 1]

22 Sep 2020

PONE-D-20-07225R1

Dietary nutrients of relative importance associated with coronary artery disease: Public health implication from random forest analysis

PLOS ONE

Dear Dr. Basnet,

Thank you for submitting your manuscript to PLOS ONE. After careful consideration, we feel that it has merit but does not fully meet PLOS ONE’s publication criteria as it currently stands. Therefore, we invite you to submit a revised version of the manuscript that addresses the points raised during the review process.

Please accommodate the critical comments raised by the reviewer on the generalizability of the study and risk of selection bias, specially in relation with the nature of the controls. Please make sure that the major methodological limitations of the study are adequately discussed.Please explain how the values for PUFA and SFA intake considerably changed in the revised version as compared to the original one.

We look forward to receiving your revised manuscript.

Kind regards,

Samson Gebremedhin, PhD

Academic Editor

PLOS ONE

Reviewers' comments:

Reviewer's Responses to Questions

**Comments to the Author**

1. If the authors have adequately addressed your comments raised in a previous round of review and you feel that this manuscript is now acceptable for publication, you may indicate that here to bypass the “Comments to the Author” section, enter your conflict of interest statement in the “Confidential to Editor” section, and submit your "Accept" recommendation.

Reviewer #2: (No Response)

2. Is the manuscript technically sound, and do the data support the conclusions?

Reviewer #2: No

3. Has the statistical analysis been performed appropriately and rigorously? 

Reviewer #2: No

4. Have the authors made all data underlying the findings in their manuscript fully available?

Reviewer #2: Yes

5. Is the manuscript presented in an intelligible fashion and written in standard English?

Reviewer #2: Yes

6. Review Comments to the Author

Reviewer #2: First, I must thank the authors for their careful response to the previous round of reviews. Indeed, the changes the authors have made to the manuscript represent an important improvement, particularly the added details related to methodology and the supplementary data and script. However, there are still issues to be addressed:

1. Study design and generalisability

My primary concern remains the conclusions that the authors draw from their analysis. Previously, I highlighted:

“The authors have a case-matched control group – but the control group are not healthy individuals, but patients with other health conditions and cardiovascular symptoms. This is clearly evident by the way the authors chose to enrol patients – all patients were being examined because of suspected coronary problems. This makes it impossible to talk of differences between these groups in terms of risk factors. If this study is to be published, the authors need to somehow explain why this control group can be considered representative of a broader population. Later in the paper the authors do highlight the results may have been biased due to “the selection of the control group from the outdoor patients from the same hospital where more hypertensive patients come for their heart check-up”, but this seems to critically undermine the entire results of this study.”

This remains my position. The authors have not adjusted their conclusions or their interpretation of their study in light of the problems related to their study design. There is no attempt in this study to understand whether the control group is representative of the broader Nepalese population, and there is a major risk of selection bias. This is clearly indicated by the similar number of hypertensive patients in the case and control groups. For example, the major conclusion of this paper remains:

“Thus, a dietary intervention approach in CVDs is an effective strategy to reduce the public health burden. We conclude that dietary SFA, vitamin A.R.E., dietary total fat and oil, β-carotene, and cholesterol are topmost five essential dietary nutrients associated with CAD in the Nepalese population.”

The findings of this paper cannot be extended to make claims about the relationship between the intake of any dietary nutrient in the wider Nepalese population and their risk of CAD. The findings are at best suggestive of a possible relationship between these nutrients and the development of CAD, but prospective cohort studies and RCTs will need to be performed, as the authors do go on to highlight.

Further, it is also clear that the case group here differs drastically from the control group in many important aspects. Compared to the controls, the case group has more than double the number of patients with diabetes, double the number obese (BMI) patients, triple the number of patients suffering from dyslipidaemia, and triple the number who are current smokers. They cases also drink more alcohol and exercise less than the control group. These groups are not comparable – and clearly have very different lifestyles, so it is not clear to me that the authors can draw any conclusions about the respective role of specific nutrients in explaining CAD between the groups.

Accordingly, more needs to be done to modify the conclusions of this paper and highlight the limitations of this study before publication. As there are questions over the generalisability of these findings beyond the study population, my recommendation would be to limit all conclusions to describing the findings of this study in relation to this group alone.

2. Unexplained differences in findings reported between the original and revision.

The authors must explain why some figures in this revised manuscript compared to the original have changed. Specifically, I refer to Table 2. In the original, for Food energy (kcal) per day, the controls are reported as 2560 (2306, 2791) and the cases 2549 (2256, 2897) kcal. However, in the revised manuscript, the controls now are reported as 2674 (2445, 2909) and the cases 2622 (2373, 2963). Despite this change, none of the other macronutrient values have changed, which cannot be true.

Further, I cannot understand how the authors have also changed values for PUFA and SFA intake in this revised version compared to the original – but somehow this has not changed the value for total fat intake?

In this original paper,

Fat g Control 56 (47, 64) | Case 61 (52, 72)

PUFA g Control 18 (11.2, 22.7) | Case 19.6 (11.8, 25.7)

SFA g Control 15.8 (10.8, 20)| Case 16.6 (11, 21.6)

In the revised manuscript,

Total fat/oil g Control 56 (47, 64)| Case 61 (52, 72)

PUFA g Control 18.7 (12.5, 23.5) | Case 19.6 (12.4, 25.7)

SFA g Control 15.5 (10.6, 19.2) | Case 19 (13.9, 23.6)

The authors need to explain why this discrepancy has occurred because it undermines my confidence in the reliability of these data. I am particularly worried about the change in SFA values because the authors now report a significant OR of 1.2 (1.11, 1.31) for SFA intake in the revised manuscript, which was not included in the original.

Another major change is the figures concerning vitamin A,E,R intake between these versions:

In the original:

Vitamin A R.E. Control 739 (578, 885)| Case 657 (535, 790)

In the revised:

Vitamin A R.E. Control 698 (546, 836)| Case 622 (506, 728)

Either there was a major problem with the basic statistical analysis performed for the previous version of this paper, or these data have since been changed.

3. Referencing problems

Some references still do not clearly relate to the statements made by the authors. For example, the authors state:

"An estimated 7.4 million people died from CAD in 2015, representing 13% of all global deaths. In Nepal, 30% of total death was related to cardiovascular disease [2]".

The reference provided for this is “WHO. Cardiovascular diseases fact sheet. WHO. World Health Organization; 2017”. Nowhere in this referenced document are either of these figures provided.

4.Summary

My recommendation, despite the improvements to the paper in many areas, remains to reject this paper. Previously, this was based on the flawed study design that, I believed, undermined the generalisability of these results to the broader Nepalese population. However, the unexplained changes to nutrient intakes in this revised version undermined my confidence in this study.

7. PLOS authors have the option to publish the peer review history of their article (what does this mean?). If published, this will include your full peer review and any attached files.

Reviewer #2: No

---

## [Author Response · Author response to Decision Letter 1]

27 Oct 2020

Responses to the academic editor and the reviewer’s comments

Thank you so much for all the comments, which were valuable and encouraging to make quality manuscript publishable in the PLoS One journal. We detail our responses to each of the academic editor and the reviewers’ comments in the order they were raised. We believe that we have improved the methodological details for correct interpretation of the study.

Responses to the academic editor's comments

1. Please accommodate the critical comments raised by the reviewer on the generalizability of the study and risk of selection bias, specially in relation with the nature of the controls. 

Response: Bias arises in case-control studies not because the cases and controls differ on characteristics other than exposure but because the selected controls do not accurately reflect exposure prevalence in the study base-geographic scope, socioeconomic, and behavioral characteristics of the source population for these cases. Conceptually, the study base is populated by the people at risk of becoming identified as cases in the study if they got the disease during the time in which cases are identified. However, finding the base population in hospital control is pragmatically difficult as the conceptual definition of the study base is unclear. Moreover, the sampling procedure (unbiased sampling) in such a study is not practical. Therefore, we carried out a matched case-control study to overcome the above problems. We understand that obtaining perfectly coherent case and control groups from the same study base guarantees that there will be no additional selection bias introduced in the case-control sampling beyond whatever selection bias may be inherent in the underlying cohort. The failure to do so, however, does not automatically produce selection bias; it just introduces the possibility.

Our study aimed to determine the association of nutritional factors with coronary artery disease (CAD). The distribution of our interest variables (dietary nutrients) in the control group was within the range of findings in a previous study conducted among Nepalese [1]; thus, we believe our control group represents the base/source population. We excluded the participants who adopted dietary modification after being diagnosed with any cardiometabolic risk factors (hypertension, dyslipidemia, diabetes, obesity). The hospital where we carried out the data collection was a cardiac referral center (Only one national cardiac center in Nepal), and people come for their health check-ups from across the country. Furthermore, cardiometabolic characteristics of our control group were somehow similar to the base (source) population except for hypertension [2]. Thus, we believe the control represents the broader Nepalese population. Although controls were selected from the same hospital with other health conditions, the effect of such conditions was adjusted during statistical analysis. Moreover, in a case-control study, we believe more the characteristics (demographic, cardiometabolic, lifestyle-related factors) of cases and controls are matched except study variable (nutritional factors), more accurately we can determine the strength of association of study variables with the disease (CAD). 

2. Please make sure that the major methodological limitations of the study are adequately discussed.

Response: We have revised the major methodological limitations of the study as per reviewer suggestions, which are highlighted in the ‘Revised manuscript with Track Changes.’

3. Please explain how the values for PUFA and SFA intake considerably changed in the revised version as compared to the original one.

Response: We apologize that we did not explain the reason why some figures in the revised manuscript compared to the original has changed in the previous review. We calculated the nutrients of individual observation from the nutrients calculator developed in an excel sheet based on the value from Nepal food composition table 2017 and transferred to the datasheet. We meticulously re-check our nutrients conversion from the food list. While transferring the data, some values were missing in SFA. Therefore value in the SFA has been changed while total fat/oil intake remains the same. The same problems happened to data for food energy per day. Regarding Vitamin A.R.E, the value of Vitamin A.R.E per 100g vegetables was incorrectly placed in the nutrient calculator formula. After correcting this problem, the final value has been changed.

Responses to the reviewr's comments

1. Study design and generalizability My primary concern remains the conclusions that the authors draw from their analysis. Previously, I highlighted:

“The authors have a case-matched control group – but the control group are not healthy individuals, but patients with other health conditions and cardiovascular symptoms. This is clearly evident by the way the authors chose to enrol patients – all patients were being examined because of suspected coronary problems. This makes it impossible to talk of differences between these groups in terms of risk factors. If this study is to be published, the authors need to somehow explain why this control group can be considered representative of a broader population. Later in the paper the authors do highlight the results may have been biased due to “the selection of the control group from the outdoor patients from the same hospital where more hypertensive patients come for their heart check-up”, but this seems to critically undermine the entire results of this study.”

This remains my position. The authors have not adjusted their conclusions or their interpretation of their study in light of the problems related to their study design. There is no attempt in this study to understand whether the control group is representative of the broader Nepalese population, and there is a major risk of selection bias. This is clearly indicated by the similar number of hypertensive patients in the case and control groups. For example, the major conclusion of this paper remains:

Response: Bias arises in case-control studies not because the cases and controls differ on characteristics other than exposure but because the selected controls do not accurately reflect exposure prevalence in the study base-geographic scope, socioeconomic, and behavioral characteristics of the source population for these cases. Conceptually, the study base is populated by the people at risk of becoming identified as cases in the study if they got the disease during the time in which cases are identified. However, finding the base population in hospital control is pragmatically difficult as the conceptual definition of the study base is unclear. Moreover, the sampling procedure (unbiased sampling) in such a study is not practical. Therefore, we carried out a matched case-control study to overcome the above problems. We understand that obtaining perfectly coherent case and control groups from the same study base guarantees that there will be no additional selection bias introduced in the case–control sampling beyond whatever selection bias may be inherent in the underlying cohort. The failure to do so, however, does not automatically produce selection bias; it just introduces the possibility.

Our study aimed to determine the association of nutritional factors with coronary artery disease (CAD). The distribution of our interest variables (dietary nutrients) in the control group was within the range of findings in a previous study conducted among Nepalese [1]; thus, we believe our control group represents the base/source population. We excluded the participants who adopted dietary modification after being diagnosed with any cardiometabolic risk factors (hypertension, dyslipidemia, diabetes, obesity). The hospital where we carried out the data collection was a cardiac referral center (Only one national cardiac center in Nepal), and people come for their health check-ups from across the country. Furthermore, cardiometabolic characteristics of our control group were somehow similar to the base (source) population except for hypertension [2]. Thus, we believe the control represents the broader Nepalese population. Although controls were selected from the same hospital with other health conditions, the effect of such conditions was adjusted during statistical analysis. Moreover, in a case-control study, we believe more the characteristics (demographic, cardiometabolic, lifestyle-related factors) of cases and controls are matched except study variable (nutritional factors), more accurately we can determine the strength of association of study variables with the disease (CAD). 

 “Thus, a dietary intervention approach in CVDs is an effective strategy to reduce the public health burden. We conclude that dietary SFA, vitamin A.R.E., dietary total fat and oil, β-carotene, and cholesterol are topmost five essential dietary nutrients associated with CAD in the Nepalese population.”

The findings of this paper cannot be extended to make claims about the relationship between the intake of any dietary nutrient in the wider Nepalese population and their risk of CAD. The findings are at best suggestive of a possible relationship between these nutrients and the development of CAD, but prospective cohort studies and RCTs will need to be performed, as the authors do go on to highlight.

Response: Thank you for your constructive feedback. We are pleased to work with the meticulous reviewer and have the opportunity to enhance our knowledge of science. We agreed with your recommendation that a possible relationship between these nutrients and the development of CAD. We have changed our manuscript as per your suggestion and added the statement as

 “The findings are at best suggestive of a possible relationship between these nutrients and the development of CAD, but prospective cohort studies and RCTs will need to be performed.”

Further, it is also clear that the case group here differs drastically from the control group in many important aspects. Compared to the controls, the case group has more than double the number of patients with diabetes, double the number obese (BMI) patients, triple the number of patients suffering from dyslipidaemia, and triple the number who are current smokers. They cases also drink more alcohol and exercise less than the control group. These groups are not comparable – and clearly have very different lifestyles, so it is not clear to me that the authors can draw any conclusions about the respective role of specific nutrients in explaining CAD between the groups.

Response: If the comparison was performed only each nutrient with CAD without adjustment with the above-mentioned variables like in 2*2 table, these groups were not comparable. However, our conclusion was based on random forest analysis and multivariable conditional logistic regression, where the effects of these confounding variables were adjusted. Therefore, conclusions about the respective role of specific nutrients in CAD between the groups were justifiable.

Accordingly, more needs to be done to modify the conclusions of this paper and highlight the limitations of this study before publication. As there are questions over the generalisability of these findings beyond the study population, my recommendation would be to limit all conclusions to describing the findings of this study in relation to this group alone.

Response: We agreed with your recommendation to limit all conclusions to describing the findings of this study about this group alone, and have changed our manuscript accordingly. The revised sentences were as:

“Our study suggests higher dietary intake of β-carotene and vitamin C are possible protective dietary nutrients, while an increased intake of dietary SFA, total fat and oil, and cholesterol are potential risk factors for CAD development. However, prospective cohort and RCTs studies with a large sample size are needed to explore the causal link of these nutrients for the risk of CAD development in the Nepalese population.”

2. Unexplained differences in findings reported between the original and revision.

The authors must explain why some figures in this revised manuscript compared to the original have changed. Specifically, I refer to Table 2. In the original, for Food energy (kcal) per day, the controls are reported as 2560 (2306, 2791) and the cases 2549 (2256, 2897) kcal. However, in the revised manuscript, the controls now are reported as 2674 (2445, 2909) and the cases 2622 (2373, 2963). Despite this change, none of the other macronutrient values have changed, which cannot be true.

Further, I cannot understand how the authors have also changed values for PUFA and SFA intake in this revised version compared to the original – but somehow this has not changed the value for total fat intake?

In this original paper,

Fat g Control 56 (47, 64) | Case 61 (52, 72)

PUFA g Control 18 (11.2, 22.7) | Case 19.6 (11.8, 25.7)

SFA g Control 15.8 (10.8, 20)| Case 16.6 (11, 21.6)

In the revised manuscript,

Total fat/oil g Control 56 (47, 64)| Case 61 (52, 72)

PUFA g Control 18.7 (12.5, 23.5) | Case 19.6 (12.4, 25.7)

SFA g Control 15.5 (10.6, 19.2) | Case 19 (13.9, 23.6)

The authors need to explain why this discrepancy has occurred because it undermines my confidence in the reliability of these data. I am particularly worried about the change in SFA values because the authors now report a significant OR of 1.2 (1.11, 1.31) for SFA intake in the revised manuscript, which was not included in the original.

Response: We apologize that we did not explain why some figures in the revised manuscript compared to the original had changed in the previous review. We calculated the nutrients of individual observation from the nutrients calculator developed in an excel sheet based on the value from Nepal food composition table 2017 and transferred to the datasheet. We meticulously re-check our nutrients conversion from the food list. While transferring the data, some values were missing in SFA. Therefore value in the SFA has been changed while total fat/oil intake remains the same. The same problems happened to data for food energy per day.

Another major change is the figures concerning vitamin A,E,R intake between these versions:

In the original:

Vitamin A R.E. Control 739 (578, 885)| Case 657 (535, 790)

In the revised:

Vitamin A R.E. Control 698 (546, 836)| Case 622 (506, 728)

Either there was a major problem with the basic statistical analysis performed for the previous version of this paper, or these data have since been changed.

Response: Regarding Vitamin A.R.E, the value of Vitamin A.R.E per 100g vegetables was incorrectly placed in the nutrient calculator formula. After correcting this problem, the final value has been changed.

3. Referencing problems

Some references still do not clearly relate to the statements made by the authors. For example, the authors state:

"An estimated 7.4 million people died from CAD in 2015, representing 13% of all global deaths. In Nepal, 30% of total death was related to cardiovascular disease [2]".

The reference provided for this is “WHO. Cardiovascular diseases fact sheet. WHO. World Health Organization; 2017”. Nowhere in this referenced document are either of these figures provided.

Response: We have cited it from the WHO webpage and changed to the correct citation for that webpage as [3]. Also we removed the statement, “In Nepal, 30% of total death was related to cardiovascular disease” from that sentence in the manuscript.

4. Summary

My recommendation, despite the improvements to the paper in many areas, remains to reject this paper. Previously, this was based on the flawed study design that, I believed, undermined the generalisability of these results to the broader Nepalese population. However, the unexplained changes to nutrient intakes in this revised version undermined my confidence in this study.

Response: We have addressed the issue raised and revised our manuscript as per reviewer’s suggestions regarding the generalizability of the results. We have meticulously re-checked each nutrient’s value so that current values differ from the original manuscript (first version). We again apologize for the reason for changes that were not mentioned in the previous review. Finally, we have changed the manuscript as per your advice (an expert reviewer) that we believe enough quality for publication in the current journal.

References

1. Shrestha A, Koju RP, Beresford SAA, Chan KCG, Connell FA, Karmacharya BM, et al. Reproducibility and relative validity of food group intake in a food frequency questionnaire developed for Nepalese diet. International journal of food sciences and nutrition. 2017;68(5):605-12. Epub 2017/01/18. doi: 10.1080/09637486.2016.1268099. PubMed PMID: 28092991.

2. NHRC. Nepal Steps Survey 2019-Facts Sheets-Nepal Health Research Council. [retrieved 2020-10-06]; Available from: http://nhrc.gov.np/nepal-steps-survey-2019-fact-sheets/.

3. WHO. Cardiovascular diseases fact sheet. World Health Organization; 2017. [retrieved 2020-7-30]; Available from: https://www.who.int/en/news-room/fact-sheets/detail/cardiovascular-diseases-(cvds).

---

## [Editor Report · Decision Letter 2]

4 Nov 2020

PONE-D-20-07225R2

Dietary nutrients of relative importance associated with coronary artery disease: Public health implication from random forest analysis

PLOS ONE

Dear Dr. Basnet,

Thank you for submitting your manuscript to PLOS ONE. After careful consideration, we feel that it has merit but does not fully meet PLOS ONE’s publication criteria as it currently stands. Therefore, we invite you to submit a revised version of the manuscript that addresses the points raised during the review process.

I once again recommend the authors to recheck their analysis on the comparison the two groups based on age. Having the same median and inter-quartile range, how it would be possible to have statistically very significant difference between the two groups? Please check the analysis again.In the abstract section please clearly provide the definitions used for “cases” and “controls”.Please describe how you identified participants who adopted dietary modification after being diagnosed with any cardiometabolic risk factors.Please provide the food frequency questionnaire (FFQ) used in the study as a supporting file with the manuscript.Regarding your request for change of authorship, please describe the contribution of the excluded authors based on the four authorship criteria stated in the PLOS ONE authorship policy.

We look forward to receiving your revised manuscript.

Kind regards,

Samson Gebremedhin, PhD

Academic Editor

PLOS ONE

---

## [Author Response · Author response to Decision Letter 2]

12 Nov 2020

Responses to the academic editor’s comments

Thank you so much for all the comments, which were valuable and encouraging to make quality manuscript publishable in the world-famous journal, PLoS One. We have revised the manuscript addressing the quarries raised. 

• I once again recommend the authors to recheck their analysis on the comparison the two groups based on age. Having the same median and inter-quartile range, how it would be possible to have statistically very significant difference between the two groups? Please check the analysis again.

Response: Thank you for your suggestion for checking the analysis regarding “Age” variable again. While re-checking, we executed four analyses of the same variable (Age): t-test (parametric and independent data), paired t-test (paired and parametric data), and Wilcoxon Rank-Sum test (independent and non-parametric data), and Wilcoxon Signed-rank test (paired and non-parametric data). The p-values for each analysis were 0.869, 0.0002, 0.893, and 0.00016, respectively. As the editor raised queries that how it would be possible to have a statistically significant difference to the data set having the same median and inter-quartile range, we did not observe statistically significant differences in the third analysis (p=0.893 ). However, the last analysis, the Wilcoxon Signed-rank test, was the appropriate analysis for our data set, which were paired (matched case-control) and non-parametric data. If we exactly matched the “Age” variable individually, it would be expected non-significance statistically. This statistical significance in the present analysis was due to the interval matching of the “Age” variable. 

• In the abstract section please clearly provide the definitions used for “cases” and “controls”.

Response: We appreciate your advice to add the definition of “case” and “control” in the abstract section. We added as:

In the present study, patients with more than seventy percent stenosis in any main coronary artery branch in angiography were defined as cases, while those presenting normal coronary angiography or negative for stressed exercise test were considered controls.

• Please describe how you identified participants who adopted dietary modification after being diagnosed with any cardiometabolic risk factors.

Response: We reviewed the patient’s record file of those diagnosed as having coronary artery disease, potentially eligible for “case” in our study. For patients taking anti-diabetic or anti-dyslipidemic or anti-hypertensive medicine for more than one month, we asked them for any modification of their usual diet. If they reported modification of their usual diet, we did not interview and exclude them from the study.

• Please provide the food frequency questionnaire (FFQ) used in the study as a supporting file with the manuscript.

Response: We divided the questionnaire into three parts, namely i. General socio-demographic characteristic ii. Cardio-metabolic and behavioral factors and iii. Food frequency questionnaire (FFQ). The FFQ was a part of the questionnaire set, which was submitted as a supplementary file S3 file. 

• Regarding your request for change of authorship, please describe the contribution of the excluded authors based on the four authorship criteria stated in the PLOS ONE authorship policy.

Response: Ali Asghar Mirzat, Falak Zeb, and Wiwik Indayati contributed revising and approving the first version of the manuscript. Mohammed Lamin Sambou supported preliminary data analysis, revision, and approval of the manuscript’s first version.

---

## [Editor Report · Decision Letter 3]

16 Nov 2020

Dietary nutrients of relative importance associated with coronary artery disease: Public health implication from random forest analysis

PONE-D-20-07225R3

Dear Dr. Basnet,

We’re pleased to inform you that your manuscript has been judged scientifically suitable for publication and will be formally accepted for publication once it meets all outstanding technical requirements.

Kind regards,

Samson Gebremedhin, PhD

Academic Editor

PLOS ONE
---

## [Editor Report · Acceptance letter]

2 Dec 2020

PONE-D-20-07225R3 

Dietary nutrients of relative importance associated with coronary artery disease: Public health implication from random forest analysis 

Dear Dr. Basnet:

I'm pleased to inform you that your manuscript has been deemed suitable for publication in PLOS ONE. Congratulations! Your manuscript is now with our production department. 

Kind regards, 

on behalf of

Dr. Samson Gebremedhin 

Academic Editor

PLOS ONE